

# Assessing potential storm tide inundation hazard under climate change: a case study of Southeast China coast

Bingchuan Nie[1,2], Qingyong Wuxi[3,4], Jiachun Li[3,4], and Feng Xu[1,2]

[1]School of Civil Engineering, Beijing Jiaotong University, Beijing 100044, China
[2]Beijing's Key Laboratory of Structural Wind Engineering and Urban Wind Environment, Beijing 100044, China
[3]Key Laboratory for Mechanics in Fluid Solid Coupling Systems, Institute of Mechanics, Chinese Academy of Sciences, Beijing 100190, China
[4]School of Engineering Science, University of Chinese Academy of Sciences, Beijing 100049, China

**Correspondence:** Jiachun Li (jcli08@imech.ac.cn)

**Abstract.** A methodology for assessing the storm tide inundation under TCI (tropical cyclone intensification) and SLR (sea level rise) is proposed, which integrates the trend analysis, numerical analysis and GIS-based analysis. In the trend analysis, the potential TCI and SLR can be estimated based on the long-term historical data of TC (tropical cyclone) and MSL (mean sea level) considering the non-stationary and spatially non-uniform effect; the numerical simulation is relied on the

ADCIRC+SWAN model, which is capable of taking into account the tide-surge-wave coupling effect to improve the precision of water elevation prediction; the water elevation is then analyzed on the GIS platform, the potential inundation regions can be identified. Based on this methodology, a case study for the Southeast China coast, one of the storm surge prone areas in China, is presented. The results show that the high water elevation tends to occur in the bays and around the estuaries, the maximal water elevations caused by the typhoon wind of 100-year recurrence period can reach as high as 6.06 m, 5.82 m and 5.67 m

around Aojiang, Feiyunjiang and Oujiang river estuaries, respectively. Non-stationary TCI and SLR due to climate change can further deteriorate the situation and enhance the risk of inundation there, i.e. the potential inundation area would expand by 108% to about 798 km$^2$ compared with the situation without considering TCI and SLR. In addition, the remotely sensed maps and inundation durations of the hardest hit regions are provided, which will aid the prevention and mitigation of storm tide inundation hazard and future coastal management there.

## 1 Introduction

Storm surge caused by tropical cyclone (TC) is one of the most hazardous events for coastal zones. Superimposing over the astronomical tide, it has produced devastating damage in low-lying areas worldwide in history (Li et al., 2017). Although the

operational forecast of storm tide (nonlinear superposition of storm surge and astronomical tide) has made a great progress



in the past decades, a few individual disastrous storm events still caused extensive damages recently, say, TC Haiyan in 2013 resulted in 6300 dead and 1061 missing (Lagmay et al., 2015).

Inundation is one of the most catastrophic consequences in a storm tide event. To reduce the potential losses during inundation, risk assessment is an effective and practical way. Based on the empirical model, Hsu et al. (2018) examined the risk
of northern Gulf of Mexico coast exposed to storm surge. Christie et al. (2018) investigated the coastal flood risk in North Norfolk, in which CS3X Continental Shelf tidal surge model and 1D SWAN model are adopted. Younus et al. (2017)analyzed the vulnerability issues during storm surge, fifty-six vulnerability issues are identified, and their categories and weighted index scale are provided. Based on a surrogate model, Taflanidis et al. (2013) carried out the risk estimation of TC waves, water elevations, and run-up for TC passing by the Island of Oahu. In those works, risk assessment is usually divided into hazard and
vulnerability assessments. The vulnerability assessment is devoted to figure out the resistant capability of coastal zones towards storm tide inundation. While, the hazard assessment, the prerequisite of vulnerability assessment, is aimed to evaluate the natural attributes of storm tide inundation. However, the precisions of water elevation prediction models in the aforementioned works are limited resulting in quite coarse hazard assessment.

Many researchers were devoted to developing hydrodynamic models with good precision and high performance for storm
tide prediction. Jelesnianski et al. (1992) built the popular SLOSH model which is still extensively used in operational forecast of storm surge. The continual renovation ADCIRC model developed by Luettich and his colleagues (Luettich et al., 1992) has been widely applied in the academic communities. Other hydrodynamic models such as POM (Peng et al., 2004), ROMS (Li et al., 2006), ELCIRC (Stamey et al., 2007), FVCOM (Weisberg et al., 2008), SELFE (Shen et al., 2009) and CH3D (Sheng et al., 2010) have been used for storm surge studies as well. The aforementioned hydrodynamic storm surge models have been
further upgraded to couple with the wave models recently. The typical works are: the ADCIRC+SWAN model (Dietrich et al., 2011, 2012), the FVCOM-SWAVE model (Wu et al., 2011)and the SELFE-WWM-II model (Roland et al., 2012). Based on those coupled models, the surge-tide-wave coupling effects can be readily presented. Zhang et al. (2017) concluded that the deviation of water elevation of order of one meter can be caused owing to nonlinear tide-surge interaction. Wuxi et al. (2018) observed wave induced surge and tide-surge interaction along Zhe-Min coast. They found that the relative error due
to tide-surge coupling effect can be as high as 28% maximal. These facts imply that the surge-tide-wave coupled models are preferred in the hazard assessment of storm tide inundation (not common at present).

In addition to the precision of hydrodynamic model, the challenging issues due to climate change need to be considered for long-term hazard assessment. Tropical cyclone intensification (TCI) and sea level rise (SLR) are obvious among the direct influential factors. Webster et al. (2005)examined the intensity of TC. They pointed out that TCs in the strongest Saffir-Simpson
categories 4 and 5 have almost doubled in both number and proportion in the period 1970-2004 for all ocean basins. Relied on the satellite-based estimation of TC intensity, Elsner et al. (2008) reported that significant wind speed upward trends of 0.3-0.09 $ms^{-1} yr^{-1}$ can be observed for the strongest TCs. The study by Knutson et al. (2010) shows that the globally averaged intensity of TC could increase 2-11% by 2100 due to greenhouse warming. Recently, Wang et al. (2016) examined the extreme wind speeds in SCS and NWP, and announced that the spatial inhomogeneous and non-stationary effects should be considered.As
for the SLR, the AR5 of IPCC reported that the global SLR for 2081-2100 relative to 1986-2005 will likely be 0.26 to 0.55





m for RCP 2.6, 0.32 to 0.63 m for RCP 4.5, 0.33 to 0.63 m for RCP 6.0, and 0.45 to 0.82 m for RCP 8.5 (IPCC, 2015). They also claimed about 70% of the coastlines worldwide would experience lower SLR than the global mean SLR by 2100s. Those results demonstrate that both TCI and SLR have significant spatial inhomogeneous and non-stationary effects.

Recently, the impacts of TCI and SLR on storm surge in a few regions have been investigated preliminary. Feng et al. (2018)
examined the storm surge trends in the coastal areas of China from 1997 to 2016. They concluded that the increasing rate of extreme storm surge is as high as 0.06 m per year at 90% confidence level. Li et al. (2018) studied the coastal flood hazards at Oahu for 24 TCs in 2080-2099 by the CMIP5 NCAR-CCSM4 model and the SLR under Representative Concentration Pathway (RCP) 8.5. Wang et al. (2012) simulated the inundation in Shanghai considering SLR, land subsidence. They found that 46% of the seawalls and levees are projected to be overtopped by 2100. Feng et al. (2018) investigated the inundation risk
of extreme water elevations in Rongcheng based on the Pearson Type III distribution considering SLR under situations RCPs 2.6, 4.5 and 8.5. Yin et al. (2017) studied the impacts of SLR and TCI artificially designed on storm surges and waves at Pearl River Estuary. However, the TCI and SLR are usually artificial gave or use the global TCI and SLR results, which means the spatial inhomogeneous and non-stationary effects is neglected.

Motivated by the aforementioned facts, a methodology for long-term hazard assessment of storm tide inundation considering
TCI and SLR is implemented in this work. It integrates the high precision surge-tide-wave coupled model and trend analysis of the local TCI and SLR considering the spatial inhomogeneous and non-stationary effects. In addition, the inundation regions are identified using the Geographic information system (GIS). Based on that, a case study is presented for Southeast China coast. The annual average direct economic loss caused by storm tide in China is about 1.73 billion USD, and no downtrend can be observed over the last three decades. Southeast China coast is one of the storm surge prone areas in China where TCs
make landfall tending to have stronger intensity. The literature review shows that inundation caused by storm tide there is scarcely studied, let alone the long-term hazard assessment of storm tide inundation considering potential TCI and SLR. The corresponding results can provide reference for the local prevention and mitigation of storm tide inundation hazard and future coastal management.

## 2  Methodology

The main steps of hazard assessment of storm tide inundation considering TCI and SLR are summarized by the flow chart in figure 1. The details of the tread analysis, numerical analysis and GIS based analysis are described as below.

TCI and SLR are the most direct influential factors for long-term hazard assessment of storm tide inundation. Although numerous results about global mean TCI and SLR has provided in literatures, e.g. Elsner et al. (2008), Knutson et al. (2010), using those results directly would introduce significant deviation for a specific study area due to the significant spatial inhomo-
geneous effect. Therefore, TCI and SLR is estimated by analyzing the long-term historical data of the study area. To consider the non-stationary effect, the non-stationary extreme value theory and nonlinear polynomial fitting are adopted for the TCI and SLR, respectively. The basic conception of the non-stationary extreme value theory is that the statistical parameters of TC wind



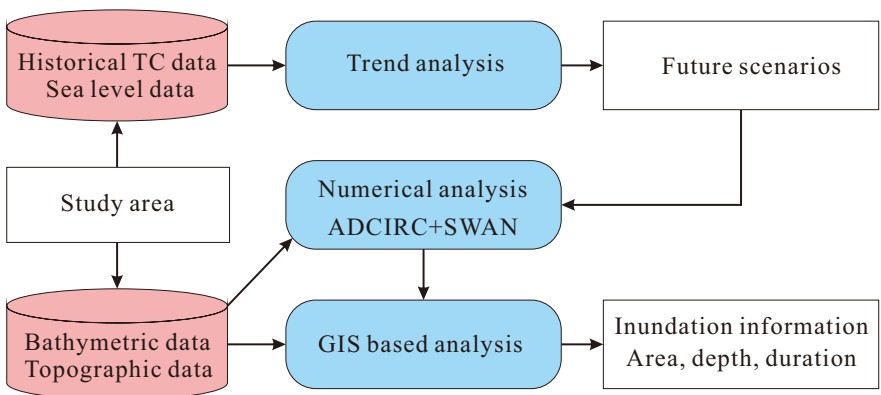

**Figure 1.** Flow chart for hazard assessment of storm tide inundation. The red cylinders and blue rounded rectangles represent the data sets and analytical approaches, respectively.

vary with time, one may refer to Wang et al. (2016). Based on the potential TCI and SLR, the parameters of the typical future scenarios can be determined.

The distribution and evolution of water elevation then can be simulated relied on the hydrodynamic model. Since surge-tide-wave coupling effects have remarkable influence on water elevation, the ADCIRC+SWAN model is adopted in this work. In ADCIRC+SWAN model, SWAN and ADCIRC are run on the same computational core and the same unstructured sub-mesh. Therefore, no interpolation is required and water elevations, currents, wind velocities and radiation stress gradients can be passed directly through local cache or memory resulting in high computational efficiency. The governing equations of

ADCIRC+SWAN model and its external forcing and dissipation strategy we used are described in Section 3.

    The numerical water elevation is then analyzed on the GIS platform, low-lying regions of inundation risk can be identified, and inundation hazard information including the area, inundation depth and duration for those regions can be figured out. Those result can provide reference for the subsequent vulnerability assessment.

## 3   Hydrodynamic surge-tide-wave coupled model

**3.1   Governing equations**

The water elevation $\zeta$ of storm tide is described by the Generalized Wave Continuity Equation (GWCE) as (1). The vertical uniform horizontal current velocity $U$ and $V$ are governed by the vertically-integrated momentum equations, i.e. (2) and (3).

$$\frac{\partial^2 \zeta}{\partial t^2} + \tau_0 \frac{\partial \zeta}{\partial t} + S_p \frac{\partial J_\lambda}{\partial \lambda} + \frac{\partial J_\varphi}{\partial \varphi} - S_p U H \frac{\partial \tau_0}{\partial \lambda} - V H \frac{\partial \tau_0}{\partial \varphi} = 0, \tag{1}$$




$$\frac{\partial U}{\partial t} + S_p U \frac{\partial U}{\partial \lambda} + V \frac{\partial U}{\partial \varphi} + \frac{\partial J_\varphi}{\partial \varphi} - fV = -g S_p \frac{\partial}{\partial \lambda}\left[\zeta + \frac{P_S}{g\rho_0} - \alpha\eta\right] + \frac{\tau_{s\lambda,winds} + \tau_{s\lambda,waves} - \tau_{b\lambda}}{\rho_0 H} + \frac{M_\lambda - D_\lambda}{H}, \tag{2}$$

$$\frac{\partial V}{\partial t} + S_p U \frac{\partial V}{\partial \lambda} + V \frac{\partial V}{\partial \varphi} + \frac{\partial J_\varphi}{\partial \varphi} - fU = -g S_p \frac{\partial}{\partial \varphi}\left[\zeta + \frac{P_S}{g\rho_0} - \alpha\eta\right] + \frac{\tau_{s\varphi,winds} + \tau_{s\varphi,waves} - \tau_{b\varphi}}{\rho_0 H} + \frac{M_\varphi - D_\varphi}{H}. \tag{3}$$

Where $\lambda$ and $\varphi$ are the longitude and latitude, respectively. $H$ is the sum of water elevation and bathymetric depth. $M_\lambda$ and $M_\varphi$ are the vertically-integrated lateral stress gradients. $D_\lambda$ and $D_\varphi$ are the momentum dispersion terms. $S_P$ is a spherical

coordinate conversion factor. $\tau_{s\lambda,winds}$ (or $\tau_{s\varphi,winds}$) and $\tau_{b\lambda}$ (or $\tau_{b\varphi}$) are the winds and bottom friction stresses, respectively.

$J_\lambda$ and $J_\varphi$ in (1) to (3) can be rewritten as (4) and (5), respectively (Dietrich et al., 2011).

$$J_\lambda = - S_p Q_\lambda \frac{\partial U}{\partial \lambda} - Q_\varphi \frac{\partial U}{\partial \varphi} + f Q_\lambda - \frac{g}{2} S_p \frac{\partial \zeta^2}{\partial \lambda} - g S_p H \frac{\partial}{\partial \lambda}\left[\frac{P_s}{g\rho_0} - \alpha\eta\right]$$
$$+ \frac{\tau_{s\lambda,winds} + \tau_{s\lambda,waves} - \tau_{b\lambda}}{\rho_0} + (M_\lambda - D_\lambda) + U\frac{\partial \zeta}{\partial t} + \tau_0 Q_\lambda - g S_p H \frac{\partial \zeta}{\partial \lambda}, \tag{4}$$

$$J_\varphi = - S_p Q_\lambda \frac{\partial V}{\partial \lambda} - Q_\varphi \frac{\partial V}{\partial \varphi} - f Q_\lambda - \frac{g}{2} \frac{\partial \zeta^2}{\partial \varphi} - g H \frac{\partial}{\partial \varphi}\left[\frac{P_s}{g\rho_0} - \alpha\eta\right]$$
$$+ \frac{\tau_{s\varphi,winds} + \tau_{s\varphi,waves} - \tau_{b\varphi}}{\rho_0} + (M_\varphi - D_\varphi) + V\frac{\partial \zeta}{\partial t} + \tau_0 Q_\varphi - g H \frac{\partial \zeta}{\partial \varphi}. \tag{5}$$

Here, $g$, $\alpha$, $\rho_0$ and $\eta$ are the gravitational acceleration, the tidal potential, water density and the effective earth elasticity factor, respectively. $Q_\lambda$ (or $Q_\varphi$) is the product of current velocity and $H$.

The wave action density $N$ is conserved during propagation in the presence of ambient current. Hence, the wind wave satisfies the balance equation of the wave action density given as

$$\frac{\partial N}{\partial t} + \frac{\partial}{\partial \lambda}\left[(c_\lambda + U)N\right] + \cos^{-1}\varphi \frac{\partial}{\partial \varphi}\left[(c_\varphi + V)N\cos\varphi\right] + \frac{\partial}{\partial \theta}[c_\theta N] + \frac{\partial}{\partial \sigma}[c_\sigma N] = \frac{S_{tot}}{\sigma}. \tag{6}$$

Where, $c_\lambda$ (or $c_\varphi$) is the group velocity of water wave. $c_\sigma$ and $c_\theta$ are the propagation speeds in spectral space. The source term $S_{tot}$ is usually constituted of six processes which can be as written as (7) (Zijema et al., 2010). $S_{in}$ is the wave energy inputted by the wind. $S_{ds,w}$, $S_{ds,b}$ and $S_{ds,br}$ are wave dissipation terms caused by white capping, bottom friction and depth-induced wave breaking, respectively. $S_{nl,3}$ and $S_{nl,4}$ represent the three-wave and four-wave interactions, respectively.

$$S_{tot} = S_{in} + S_{ds,w} + S_{ds,b} + S_{ds,br} + S_{nl3} + S_{nl4}. \tag{7}$$

The parameterizations of wave energy input/dissipation terms, i.e. $S_{in}$, $S_{ds,w}$, $S_{ds,b}$ and $S_{ds,br}$, are expressed as (8) to (11). Where, $A_{in}$ and $B_{in}$ satisfy the linear growth formula (Cavaleri et al., 1981) and exponential growth formula (Komen et al., 1984), respectively. $\Gamma$ is a coefficient dependent on water depth. $C_b$ is the bottom friction coefficient. $E_{tot}$ and $D_{tot}$ are the total wave energy and the rate of dissipation due to wave breaking, respectively.

$$S_{in} = A_{in} + B_{in}E(\sigma,\theta). \tag{8}$$




$$S_{ds,w} = -\Gamma \bar{\sigma} \frac{k}{\bar{k}} E(\sigma, \theta). \tag{9}$$

$$S_{ds,b} = -C_b \frac{\sigma^2}{g^2 \sinh^2 kH} E(\sigma, \theta). \tag{10}$$

$$S_{ds,br} = \frac{D_{tot}}{E_{tot}} E(\sigma, \theta). \tag{11}$$

The coupling manner between the wave and water elevation is that: wind wave contributes to the water elevation via wave radiation stresses, i.e. $\tau_{s\lambda,winds}$ and $\tau_{s\varphi,winds}$ in (2)-(5); while the water elevation affects the wave dissipation terms (8)-(11) in turn.

In this study, the ADCIRC+SWAN model developed by Dietrich et al. (2011, 2012) is used to solve the aforementioned
coupled governing equations. The continuous-Galerkin, finite-element model ADCIRC solves (1)-(5) to obtain the time dependent water elevation and current (Jelesnianski et al., 1992). The third-generation wave model SWAN estimates the evolution of wave parameters by (6)-(11) (Booij et al., 1999). In every interval, ADCIRC accesses the gradients of radiation stresses by SWAN, while SWAN accesses the current and water elevation provided by ADCIRC.

### 3.2   External forcing and dissipation

The hydrodynamic surge-tide-wave model described by (1) to (11) is driven by the stresses due to $\tau_{s\lambda,winds}$ ($\tau_{s\varphi,winds}$) and bottom friction $\tau_{b\lambda}$ ($\tau_{b\varphi}$), the pressure $P_s$, the water elevation and current of astronomic tide. They are described as follows.

The wind stress $\tau_{s\lambda,winds}$ ($\tau_{s\varphi,winds}$) is determined by the drag coefficient $C_d$ and the wind field $V_{wind}$ at 10 m above sea surface. Here, $C_d$ increasing linearly with wind speed as proposed by Garrat (1977) is used. The wind field $V_{wind}$ is reconstructed based on the Holland model (Hubbert et al., 1991; Jakobsen et al., 2004), which can be decomposed into tangential
wind velocity $V_T$, radial wind velocity $V_r$ and environmental scale wind velocity $V_E$, see (12).

$$V_{wind} = V_T + V_r + V_E. \tag{12}$$

The expressions of $V_T$, $V_r$ and $V_E$ are given by (13)-(15), respectively.

$$V_T = \frac{1}{V_m}(\sqrt{R^{-B}\exp(1-R^{-B}) + a^2 R^2} - aR). \tag{13}$$

$$V_r = \frac{B[BR^{-2B} + (1-3B)R^{-B} + B - 1]k - 2k - RC}{B(R^{-B} - 1) + 2 + 4aR(V_T/V_m)^{-1}} V_T. \tag{14}$$

$$V_E = U_0 \exp\left(-\frac{r}{R_G}\right). \tag{15}$$

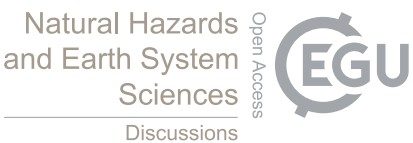

Here, $R$ equals to $r/R_M$, $r$ the distance to the TC eye. $V_m$, $U_0$, $R_G$ and $R_M$ are the maximal wind velocity, the migration speed, the length scale and the radius of maximum wind of TC, respectively. $B$, $C$ and $k$ are coefficients, which equals to 1.5, 0.013 and 0.16, respectively. $a$ demotes $fR_M/(2V_m)$, where $f$ the Coriolis parameter.

The pressure field is calculated as below (Hubbert et al., 1991; Jakobsen et al., 2004).

$$P = P_c + (P_n - P_c)\exp(-R^{-B}). \tag{16}$$

Once the maximal wind speed $V_m$, the central pressure $P_c$, the ambient pressure $P_n$ and the track of TC are given, the wind stresses $\tau_{s\lambda,winds}$ and $\tau_{s\varphi,winds}$ and the pressure $P_s$ can be reconstructed based on (12)-(16).

The bottom friction stress $\tau_{b\lambda}$ ( $\tau_{b\varphi}$) equals the product of density $\rho_0$, $K_{slip}$ and the square of current $U^2$ ($V^2$). $K_{slip}$ is the bottom friction coefficient given by

$$K_{slip} = C_{f\min}[1 + (\frac{H_{break}}{H})^{\theta_f}]^{\gamma_f/\theta_f} (U + V)^{1/2}. \tag{17}$$

Where, $H_{break}$ is the wave break depth. $C_{fmin}$, $\theta_f$ and $\gamma_f$ are set to the recommended values, i.e. 0.0026, 10 and 1/3, respectively (Luettich et al., 2006).

Eight constituents K1, K2, M2, N2, O1, P1, Q1 and S2 from the Le Provost tidal database FES95.2 are used. The tidal elevation and depth-averaged tidal current are exerted on the open boundary.

The aforementioned parameterization has been validated for TCs Thane on the India Ocean and Saomai on the Northwest Pacific Ocean. Water elevation, significant wave height and surge are compared with in-situ data, and good agreements have been observed, for which one may refer to Wuxi et al. (2018).

## 4  A case study of Southeast China coast

### 4.1  Study area

Southeast China coast as indicated by the red rectangle in figure 2 is the area of interest. The whole computation domain is extended to the Northwest Pacific Ocean (NWP) and South China Sea (SCS) to exclude the impact of the open boundary. The GEBCO bathymetry database of 30 arc second resolution is adopted. Unstructured grids with varying density, about 105k elements, are adopted for the computational domain. The mesh scale at the area of interest is less than 100 m.

### 4.2  Local TCI and SLR

Relied on the non-stationary model (Wang et al., 2016), the TC intensities of different recurrence periods over the study area are investigated. The main steps for estimating the extreme wind speed of TC are described as below: 1) the 72 years (1945-2013) TC database from JTWC is divided into 23 periods with 50 years in each period, i.e. 1945-1994, 1946-1995, ..., 1967-2013; 2) the shape and scale parameters of Weibull distribution for the wind speed of TC affect the area of interest are estimated for

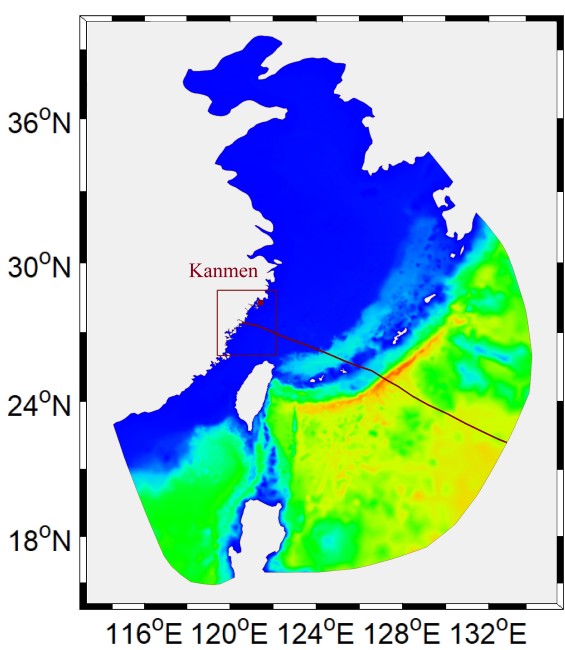

**Figure 2.** Overview of the computational domain. The region by the red rectangle is the area of interest. The coordinates of the corners of the red rectangle are (119.224 ° E, 28.687 °N), (122.049 °E, 28.687 °N), (119.224 °E,25.545 °N), (122.049 °E, 25.545 °N), respectively. The red dot (28.083 °N, 121.128 °E) is the location of Kanmen tidal gauge.

each period; 3) the time series of statistical parameters are fitted; 4) finally, the extreme wind speeds of different return periods can be obtained based on the extreme value theory as presented in figure 3. Taking the maximal wind speed of 50-year and 100-year recurrence period as examples, they can be as high as 75.49 m/s and 78.48 m/s, respectively.

Considering the significant regional differences of SLR, the observed sea level series of Kanmen tidal gauge (28.083 °N,
121.128 °E), located in the study area (see figure 2) are analyzed. The monthly and five-year averaged MSLs from 1959 to 2017 are presented in figure 4. Obvious increasing trend of SLR can be observed. Based on the in-situ data, polynomial fitting is carried out as indicated by the red solid line in figure 4. The extrapolated potential SLR for 2050s is 0.185 m relative to 2006. While that for 2100s is 0.514 m, which is between the situations of RCP 6.0 and RCP 8.5 (IPCC, 2015).

### 4.3   Scenario parameters

Saomai is the strongest TC made landfall at the study area. It struck the coast almost vertically as sketched by the solid line in figure 2. According to the National Meteorological Centre of China, the maximum wind speed of Saomai is about 60.0 m/s during landing. Therefore, scenario S2 (see table 1) is used to examine the storm tide inundation hazard without considering TCI and SLR. To evaluate the worst situation, typhoon makes landfall during the astronomical high tide in S2. Meanwhile, scenario S1, the hindcasting of actual situation of storm tide by Saomai, is also investigated. S3 and S4 as given in table 1 are

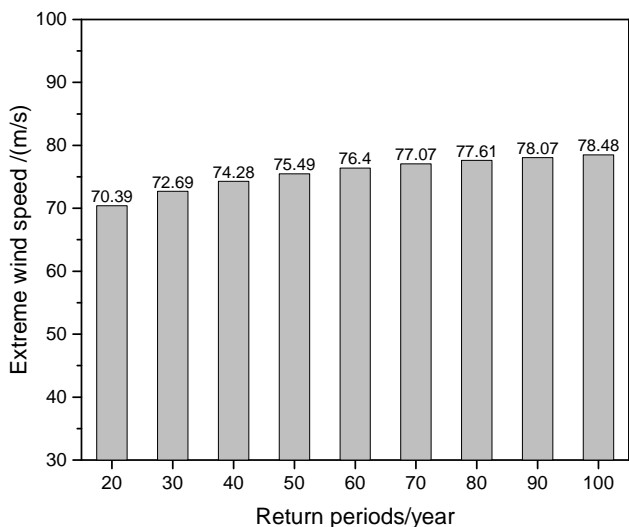

**Figure 3.** The extreme wind speed of TC with different recurrence periods for the study area considering non-stationary effect.

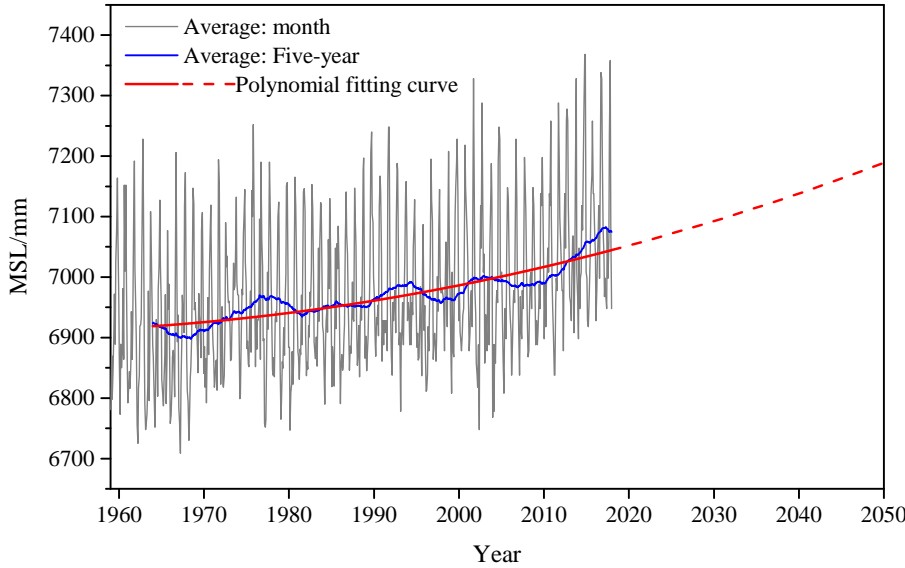

**Figure 4.** MSL series of Kanmen tidal gauge. The gray and blue solid lines are the averaged sea levels over one month and five years, respectively. The red solid line is polynomial fitting result of the five-year averaged MSL, while the red-dash line is extrapolation of the fitting result.





**Table 1.** Parameters of the scenarios concerned. Scenario S1 is the actual situation of TC Saomai. Scenario S2 is the situation that typhoon of wind speed of 100-year recurrence period makes landfall during the astronomical high tide without considering TCI and SLR. S3 (S4) corresponds to the situation that typhoon of wind speed of 50-year (100-year) recurrence period makes landfall during the astronomical high tide taking into account of the non-stationary TCI and SLR.

| Scenarios | MSL (mm) | TC intensity (m/s) | Landing moment | TC track |
|-----------|----------|--------------------|----------------|----------|
| S1 | 7013.62 | 60 | Landing moment of Saomai | Saomai |
| S2 | 7013.6 | 60.0 | High tide | Saomai |
| S3 | 7198.7 | 75.5 | High tide | Saomai |
| S4 | 7526.6 | 78.5 | High tide | Saomai |

investigated to examine the non-stationary TCI and SLR effects, respectively. According to the statistical results presented in Section 4.2, the extreme wind speeds of 50-year recurrence period and 100-year recurrence period are adopted for S3 and S4, respectively. As for TCI, SLR of 0.185 m (0.514 m) compared with 2006 is adopted for S3 (S4).

## 4.4   Water elevation

The distributions of water elevation at the landing moment for scenarios S1-S4 are presented as shown in figure 5. It indi-
cates that the distributions of water elevation for the four scenarios are similar, but with significant difference in magnitude. Comparing S1 (see figures 5(a)) and S2 (see figure 5(b)), the landing moment has dramatic impact on the magnitude of water elevation. Specifically, the maximal water elevation for S2 is as high as about 6.0 m, while that for S1 is about 3.1 m. Significant difference of maximal water elevation along coastline between S1 and S2 can also be observed in figure 7(a). It implies that the potential storm tide hazard if Saomai landed during the period of astronomical high tide would be much more catastrophic
than the actual situation of more than 480 deaths and 2.5 billion USD economic loss.

    It is noteworthy that the difference of water elevation between S2 and S1 is not a linear superposition of storm surge on different astronomical tide levels. The tide-surge-wave coupling effect also play a role. Wuxi et al. (2018)concluded that the water elevation due to coupling effect experiences insignificant premonitory fluctuation, remarkable rise/drop and residual oscillating with a gentle decay of magnitude. To observe the distribution and magnitude of impact of coupling effect on water
elevation, the difference of water elevation between S2 and S1 with the astronomical tide excluded is presented in figure 6. Difference of water elevation of -1 m maximal can be observed at area north of the TC track and in the bays. That means coupling effect can reduce the water elevation when astronomical tidal level increases. The main reasons are: 1) waves with larger height can survival in higher water depth resulting in less wave induced surge, 2) nonlinear tide-surge effect trends to reduce water elevation when astronomical tidal level increases.

The maximal water elevations along the coastline north of the landing location during the whole process of TC passing by for S1-S4 are plotted in figure 7. From figure 7, the curves of water elevation along the coastline for S2-S4 can be divided

**Figure 5.** Water elevation (m) at landing moment. (a), (b), (c) and (d) are for secnario S1, S2, S3 and S4, respectively. The dot and line are the eye and track of TC, respectively.

into three segments with obvious different magnitudes of water elevation, i.e. the segment 0-25 km, the segment 25-225 km and the segment beyond 225 km. The increase of water elevation in the segment 0-25 km is mainly induced by the longshore current blocked by the convex of the coastline when the radius of maximum wind is close to the landing location; while that in the segment 25-225 km is due to the accumulation of water body blown by wind towards coast when the eye of TC is close to the landing location; for the coastline out of the radius of maximal wind, i.e. the segment beyond 225 km, water elevation

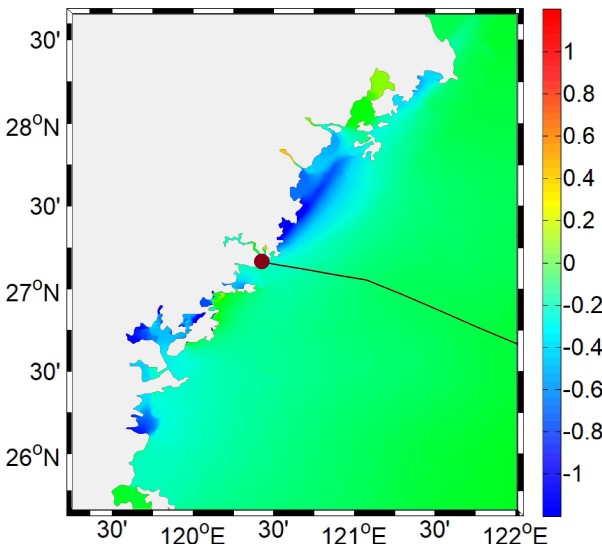

**Figure 6.** Difference of water elevation (m) between S2 and S1 with the astronomical tide excluded.

is much smaller. Figure 7 shows that the average maximal water elevations along the coastline concerned are 4.91 m, 5.25 m and 5.71 m for S2-S4, respectively. The comparison between S2 and S4 indicates that the non-stationary TCI and SLR effects increase water elevation by 0.80 m on average for the coastline concerned. More specifically, the water elevation due to SLR is

a little bit smaller than 0.51 m, while that due to TCI is a little bit larger than 0.29 m. Because the MSL of S4 is 0.51 m larger than that of S2, and the tide-surge-wave coupling effects will reduce water elevation slightly when SLR, which is similar to the behavior of the coupling effect when astronomical tidal level increases (see figure 6). Comparing S3 and S4, it shows that the non-stationary TCI and SLR effects increase water elevation along the coastline by 0.46 m averagely, in which SLR effect is likely to share about 0.33 m. Accordingly, we can conclude that water elevation will be underestimated remarkably without

considering the climate change impact, and both non-stationary TCI and SLR are important factors should be dealt with for long-term hazard assessment of storm tide.

Further, the locations of the extremums of water elevation curves in figure 7, i.e. B1-B4, are marked in figure 8. It demonstrates that high water elevation occurs in the bays or at the river estuaries, since water body is easy to accumulate in such geography. Specifically, B1 is the bay near the landing location of the TC eye; while B2-B4 are Aojiang river estuary, Feiyun-

jiang river estuary and Oujiang river estuary, respectively. The maximal water elevations at Aojiang river estuary (i.e. B2) are 6.06 m, 6.51 m and 7.02 m for S2-S4, respectively. While the maximal water elevations at Feiyunjiang river estuary (i.e. B3) are 5.82 m (S2), 6.20 m (S3) and 6.67 m (S4), respectively, and that at Oujiang river estuary (i.e. B4) are 5.67 m (S2), 6.00 m (S3) and 6.44 m (S4), respectively.


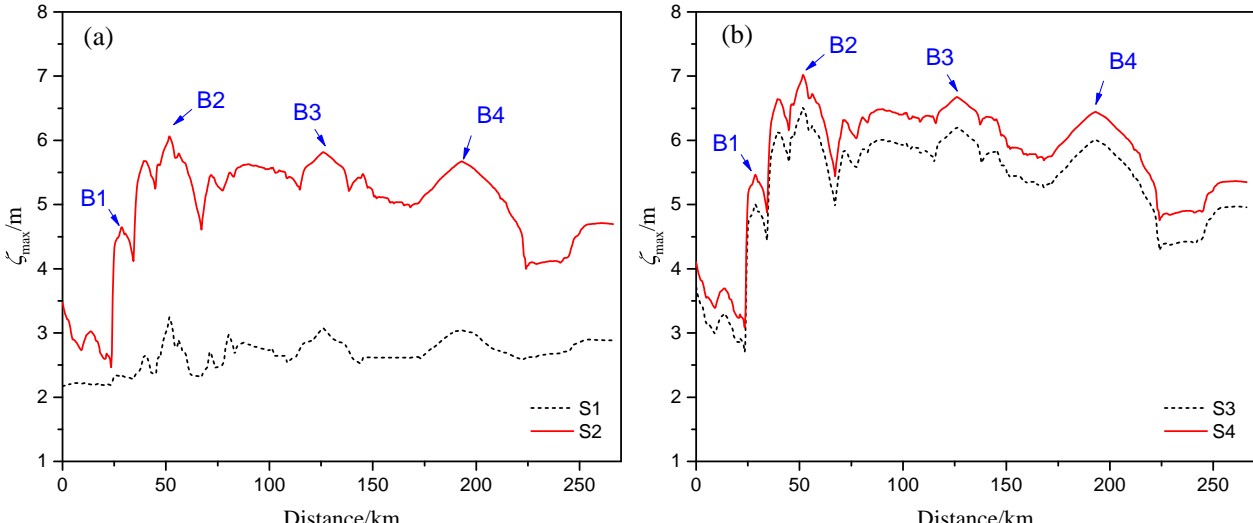

**Figure 7.** The maximal water elevation along the coastline during the whole process of TC landing. B1-B4 are the extremums of water elevation with their locations marked in figure 8. The horizontal axis represents the distance to the landing location of the TC eye.

## 4.5 Inundation

Relied on the GIS platform, the potential inundation regions with area larger than 10 km² for S2 and S4 are identified as shown in figures 8 and 9, respectively. From figure 8, the south coast of Aojiang river (R2), the south coast of Oujiang river (R5), the alluvial island at Oujiang river estuary (R6) and the north coast of Oujiang river (R7) are the largest four regions inundated. And low-lying regions R1 and R8 are also under the threat of inundation. As for S4 with the non-stationary TCI and SLR effects considered, the aforementioned six regions enlarge significantly. In addition, the north coast of Aojiang river (R3), the

south and north coasts of Feiyunjiang river (R4 and R5) expose to the storm tide as well. The areas of the aforementioned inundation regions are presented in table 2, and the lengths of the coastlines where water body intruded are also presented along with. From table 2, about 379 km² is under the threat of storm tide inundation for S2. As for S4, the potential inundation will increase by 108% to about 798 km². That is to say, the storm tide inundation area will be underestimated remarkably as well without considering the non-stationary TCI and SLR effects.

Table 2 shows that the estuaries of Aojiang river (B2), Feiyunjiang river (B3) and Oujiang river (B4) are the hardest hit regions. The remotely sensed maps of the three hardest hit regions are presented in figure 10. The coastlines where water intruded are sketched by white solid lines, the farthest boundary of water body intruded for scenarios S2 and S4 are marked by red and blue lines, respectively. Figure 10(a) shows that the Cangnan city located at the south coast of Aojiang river is suffering from storm tide inundation hazard for S2 partly. As for S4, the whole downtown of Cangnan city and almost half of

Aojiang town located at the north coast of Aojiang river are exposed to the storm tide inundation. At the estuary of Feiyunjiang river (see figure 10(b)), most of the inundation areas are farmland. Figure 10(c) shows that most of the Longwan and Dongtou
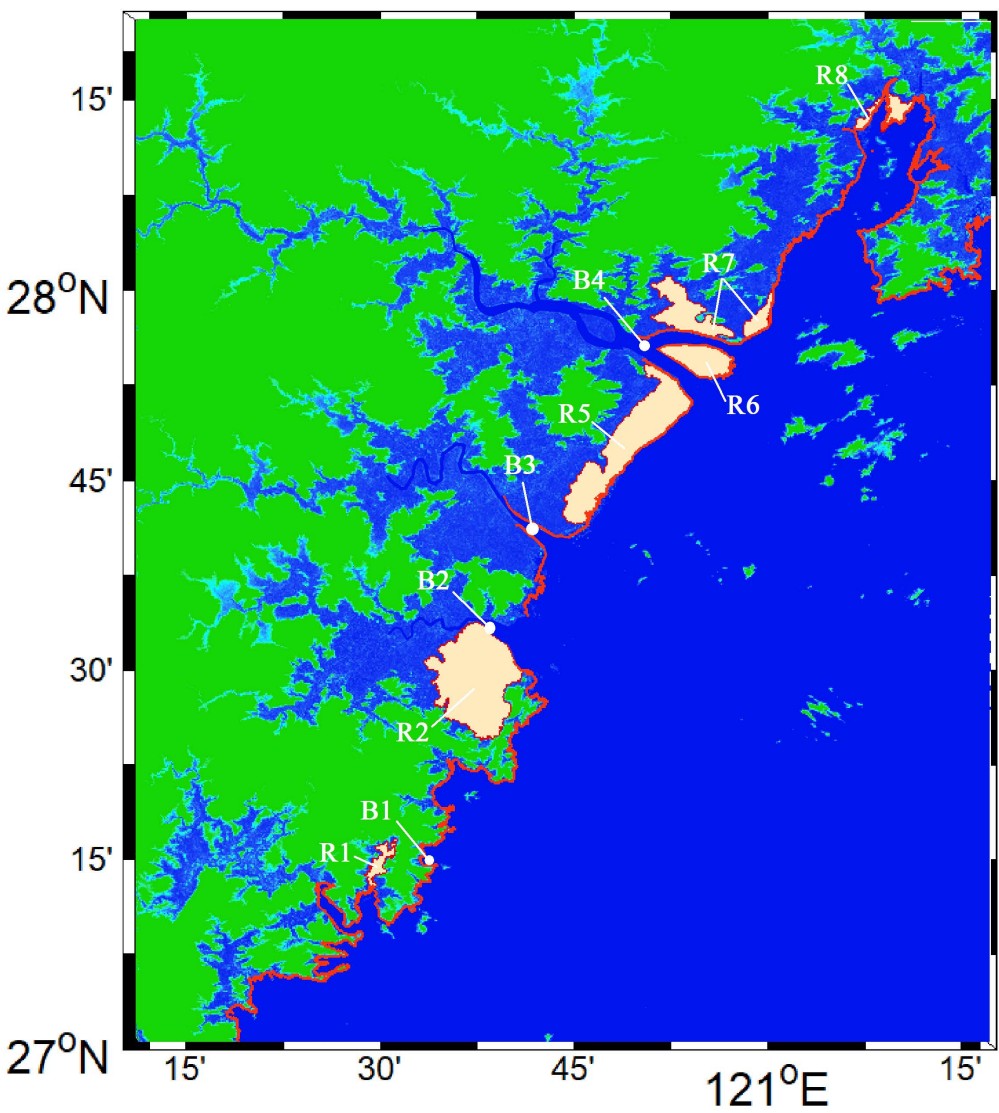

**Figure 8.** Overview of the potential inundation regions for S2. R1, R2 and R5-R8 are six regions with severe inundation hazard identified. B1-B4 are the locations of the extremums of water elevation in figure 7.

districts of Wenzhou city are of high risks of storm tide inundation. To provide reference for the local vulnerability assessments, the inundation durations for spots with different height over MSL for S2-S4 are also provided in figure 11. It shows that the inundation duration of S4 ranks the first followed by S3 and S2, which is consistent with their water elevations. Form figure


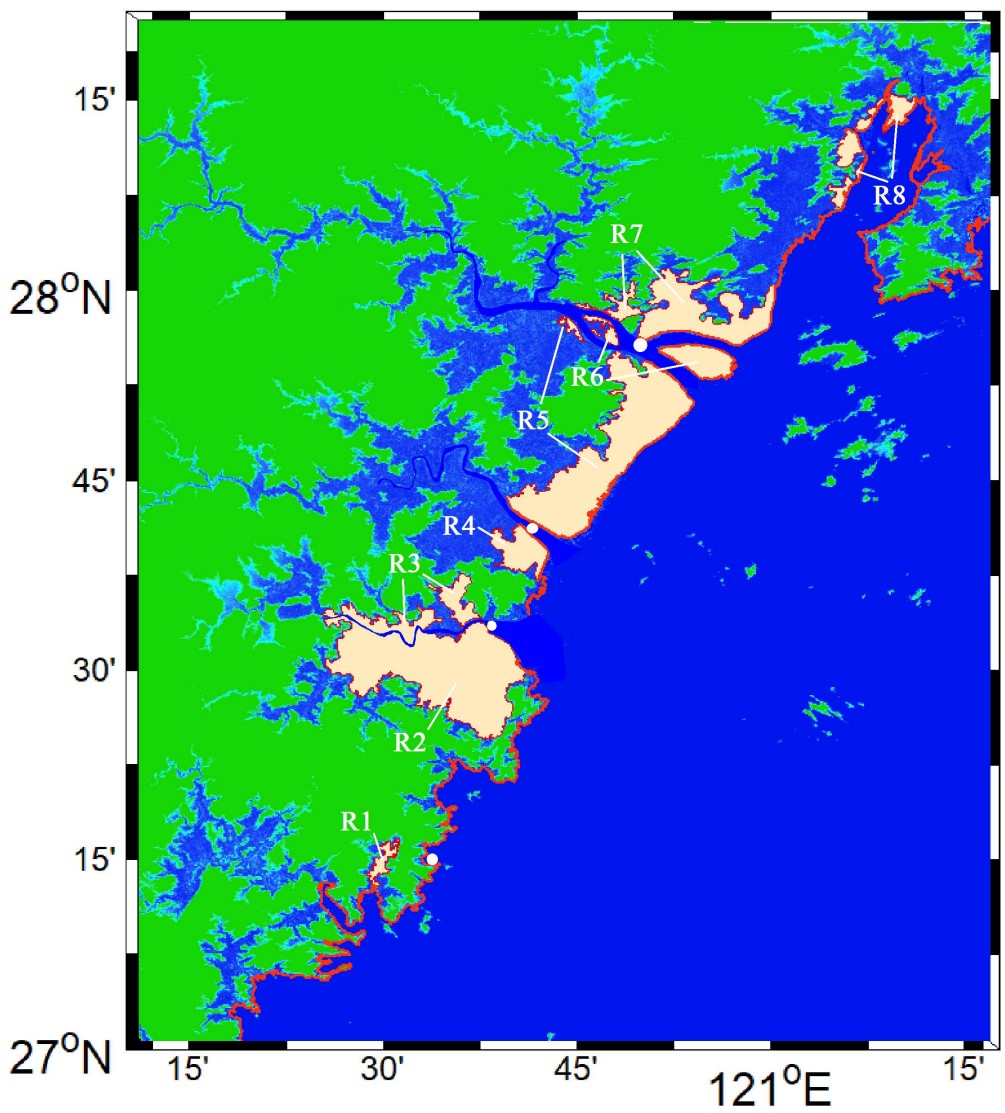

**Figure 9.** Overview of the potential inundation regions for S4. R1-R8 are eight regions with severe inundation hazard identified.

11, the inundation duration is not spatial homogeneous, and the estuary of Feiyunjiang river experiences longer period than the other two estuaries.





**Table 2.** The area and length of the coastlines where water body intruded for the inundation regions.

| Scenarios | | R1 | R2 | R3 | R4 | R5 | R6 | R7 | R8 | Total |
|---|---|---|---|---|---|---|---|---|---|---|
| S2 | Area (km$^2$) | 10.2 | 144.7 | 0 | 0 | 122.1 | 30.5 | 54.6 | 17.1 | 379.2 |
| | Length (km) | 1.6 | 13.9 | 0 | 0 | 32.8 | 25.3 | 15.5 | 17.8 | 106.9 |
| S4 | Area (km$^2$) | 10.2 | 273.9 | 47.0 | 35.3 | 235.0 | 37.6 | 117.8 | 41.5 | 789.3 |
| | Length (km) | 1.9 | 13.9 | 3.3 | 10.5 | 53.9 | 32.6 | 25.8 | 30.0 | 171.9 |

## 5 Conclusions

Storm tide is the deadliest marine hazard, which has claimed large numbers of human lives and shocking economic losses. In the future, the potential storm tide inundation is by no means optimistic due to TCI and SLR. A methodology for assessing the

storm tide inundation under TCI and SLR is designed, which includes the trend analysis, numerical analysis and GIS-based analysis. The specific procedures are: 1) analyzing the non-stationary tropical cyclone intensification (TCI) and sea level rise (SLR) statistically based on the long-term historical database; 2) examining the water elevation of typical potential storm tide event using the ADCIRC+SWAN model; 3) identifying the potential inundated regions relied on the GIS platform.

Based on this methodology, a case study focused on the storm tide inundation along Southeast China coast, one of the

storm surge prone areas in China, is carried out. The results demonstrate that the bays and estuaries in this coastal area tend to experience more dramatic rising process of water elevation. With effects of TCI and SLR neglected, the maximal water elevation around the estuary of Aojiang river by typhoon wind of 100-year recurrence period can be 6.06 m, while that around the Feiyunjiang and Oujiang river estuaries can be 5.82 m and 5.67 m, respectively. Under the same circumstances, roughly 379 km$^2$ of coast area is under the threat of storm tide inundation. Certainly, non-stationary TCI and SLR have remarkably

impacts on the storm tide elevation and subsequent inundation. The water elevations caused by typhoon wind of 50-year recurrence period and 100-year recurrence period considering TCI and SLR are provided. The maximal water elevations caused by typhoon wind of 100-year recurrence period considering TCI and SLR can be as high as 6.51 m (Aojiang river estuary), 6.20 m (Feiyunjiang river estuary) and 6.00 m (Oujiang river estuary). The corresponding potential inundated area could expand by 108% to about 798 km$^2$. The remotely sensed maps for the most heavily hit regions, i.e. the estuaries of

Aojiang, Feiyunjiang and Oujiang rivers are provided as well, which demonstrate that a few number of downtowns such as Cangnan city and Wenzhou city are exposed to the threat of storm tide inundation.

It is worth noting that the factors owing to climate change can be no longer neglected in the future risk assessment of storm tide disasters. Apart from the issues concerned in this study, relevant social and economic data are expected to be integrated in the present procedures for vulnerability assessment. Additionally, the robustness of the local system for immediate warning,

efficient evacuating and in situ conservation need to be further evaluated in the future risk analysis as well.


**Figure 10.** Remotely sensed maps of hardest hit regions of storm tide inundation. (a), (b) and (c) are for the estuaries of Aojiang river, Feiyunjiang river and Oujiang river, respectively. The white line indicates the coastline where water body intruded. The red and blue lines correspond to the inundation lines for S2 and S4, respectively.Source ©Google Earth.

*Code and data availability.* The bathymetry data is form GEBCO (General Bathymetric Chart of the Oceans) bathymetry database. Topographic data is from the SRTM (Shuttle Radar Topography Mission) database. The wind speed, central pressure and track of tropical cyclone is provided by the China Meteorological Administration. The tidal constituents are from Le Provost tidal database FES95.2. The sea level data of Kanmen tidal gauge is from the Permanent Service for Mean Sea Level (PSMSL, www.psmsl.org).



**Figure 11.** Inundation duration for the hardest hit regions of storm tide inundation. (a), (b) and (c) are for the estuaries of Aojiang river, Feiyunjiang river and Oujiang river, respectively. The horizontal axis represents the height over the MSL. The solid line, dash line and dot dash line are for scenarios S2, S3 and S4, respectively.

*Author contributions.* Bingchuan Nie contributed to the conceptualization, development of methodology and written. Wuxi Qingyong contributed to the development of methodology. Jiachun Li contributed to the conceptualization and editing. Fengxu contributed to the written and editing.

*Competing interests.* The authors declare that they have no conflict of interest.



*Acknowledgements.* This work is supported by the National Key R&D Program of China (2017YFC1404202), the National Natural Science
Foundation of China (Grant No. 11902024) and the Strategic Priority Research Programs (Category B) of the Chinese Academy of Sciences
(XDB22040203). The authors also thank Dr. Guoqin Lyu for the assistance with the non-stationary estimation of extreme wind speed of TC.



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
