# Peer review of "Assessing potential storm tide inundation hazard under climate change: a case study of Southeast China coast"

_Natural Hazards and Earth System Sciences, 2019_

## Referee Comment (RC1) · Anonymous Referee #1 · 6 Nov 2019

**General Comments**

This paper presents the results of four simulations of one typhoon (Saomai) under different scenarios: (1) the real case, (2) enhanced tide, (3) enhanced typhoon wind speed and (4) sea level rise. Simulations are performed using a coupled hydrodynamic and wave model (ADCIRC and SWAN) implemented for the Southeast China Coast. The results are presented in terms of water elevations and inundation areas near the typhoon landing location for the different scenarios. Main outcome of the paper is that with increased wind speed and sea level rise the land areas inundated are going to be larger.

[Figure]

The paper is not acceptable for publication for several different reasons: (1) the four scenarios are not well designed and based on very weak hypothesis; (2) one of the strengths of the paper would be to take advantage of the wave-current models coupling by showing waves results for the different scenarios, that are instead completely absent; (3) the paper is poorly written; (4) the results are for just one typhoon and one specific location and it is complicated to get what is the advance in science presented in this paper or the impact of the results presented.

There are several caveats in the design of the future scenarios (Sections 4.2 and 4.3). Tropical Cyclone Intensification (TCI) – this is taken into account by substituting the wind speed of the actual typhoon with the 50-year and 100-year return period wind speed. The reasoning between the choice of those return periods and the link with TCI should be better justified with supporting literature. Sea Level Rise (SLR) – the authors claim their approach in estimating sea level rise is advanced because they consider local changes in mean sea level, i.e. they extrapolate until 2050 the mean sea level from a tide gauge dataset from 1960 to present. There is an extensive literature on global and regional sea level rise projections that is overlooked in this paper. Taking the data from one tide gauge and projecting in the future with a polynomial fitting curve do not take into account that the rate of sea level rise will change in the future due to unprecedented external factors, i.e. global warming. An accurate estimate of SLR should consider the contribution to mean sea level from steric sea-level rise, dynamic sea-level change, glaciers and ice mass loss from glaciers, Greenland and Antarctic ice sheet contributions, as well as land-water storage changes and the glacial isostatic adjustment. There are regional projections available, see Jackson, Luke P., and Svetlana Jevrejeva. "A probabilistic approach to 21st century regional sea-level projections using RCP and high-end scenarios." Global and Planetary Change 146 (2016): 179-189.

Section 3.1 is not needed. ADCIRC-SWAN are standard models already described in the literature and if the authors have not changed anything in the equations, there is no

need to state them.

The paper claim they are using a wave-hydro coupled model, but then the results are presented only in terms of water elevation and inundation maps, with no mention of waves. It would have been interesting to see for example the importance of the wave-currents interactions, or at least maps of significant wave height under the different scenarios.

The significance of the results, if any, is not conveyed to the reader. The results are just for one typhoon and for one SLR level and two enhanced wind speed scenarios. The paper does not present how these results can be generalized, for example focusing on the processes, or presenting a brand new methodology.

**Specific comments**

L2. "trend analysis, numerical analysis and GIS-based analysis" – authors should be more specific

L4. "non-stationary" – it is not clear how the authors take into account the non-stationarity of tropical cyclones and mean sea level.

L9. Maximum water elevations should be compared with values during present conditions, to understand the magnitude of changes.

L22. It is worth mentioning typhoons impacting the Chinese coast, as Haynan killed 6300 people, but in the Philippines.

L23-33. This literature review does not hold together. Risk assessment, vulnerability and hazard assessment are randomly mentioned, but the previous studies are not presented in a coherent way, it is not clear what the present work is adding to previous literature.

L24. "empirical model" – authors should be more specific

L28. "surrogate model" – authors should be more specific

L32. "the precisions of water elevation prediction models in the aforementioned works are limited resulting in quite coarse hazard assessment" – this should be supported by evidence. Hazard assessment usually means to evaluate the probability/evidence of an event. Not clear what is a "coarse" hazard assessment. With a coarse resolution model?

L36. "continual renovation" – it does not sounds as the right term

L44/45 "They found . . . 28

L54. Cite Walsh, Kevin JE and McBride, John L and Klotzbach, Philip J and Balachandran, Sethurathinam and Camargo, Suzana J and Holland, Greg and Knutson, Thomas R and Kossin, James P and Lee, Tsz-cheung and Sobel, Adam and others. (2016). Tropical cyclones and climate change. Wiley Interdisciplinar Reviews: Climate Change, 7 (1), 65-89 and references therein.

L54. "NWP" - missing the acronym definition

L60/61 "increasing rate. . . level": this sentence is not clear, in 0.06 m units are not for a rate.

L67. What is the global TCI? How does it compare with the regional one the authors are using?

L73 "1.73 billion USD" – reference needed

L159. What is the length scale? From where $V_m$, $U_0$, $R_g$ and $R_m$ are taken?

L163. From where $P_c$ and $P_n$ and track of TC are taken?

L192-193. The AR5 IPCC sea level rise projections are given as the mean sea level 2081–2100 average relative to 1986–2005 average. While the authors are comparing 2006 with 2100. How is it imposed the SLR in the model?

Table 1. Scenarios S3 and S4 are not well designed, as it is impossible to understand

if the changes are due to MSL changes or wind changes.

L201. Is the maximum wind kept constant during the simulation? Or how the wind of the real typhoon is adjusted to reach the 50- and 100-year return period?

L210. Reference needed.

L212-2019. Poorly written. Not clear what "coupling effect" means and how it affects the surge (shown in Figure 6). In addition, what are the processes behind the "non-linear tide-surge effect"? High water depth (due to high tide) means reduced bottom friction and possibly increased surge. At the same time, reduced wind stress (inversely proportional to the water depth) can lead to a decrease in the surge. So, it looks like in this case the wind could be more important than the bottom friction effect since the surge reduces. The authors should explain better the effects of the different processes, including waves.

L230. Again authors should explain why it is smaller/larger.

Figure 5. The authors should use the same limits for the colorbar to help the visual comparison of the different scenarios. Figure 5 for scenario S3 and S4 is not described in the text.

L227-237. It is very difficult to understand which are the processes that are into play. Also because in scenario S3 and S4 both the wind and MSL are changing.

Figure 7. The two panels can be merged in one figure to compare better the different scenarios.

Figure 8/9. Is the red line the actual coastline? Is the blue area where wetting and drying is allowed in the model? The authors should mark the landing location of the typhoon too.

**Technical corrections** L26. (2017) analysed

L41. 2011) and

L49. 2005) examined

L54. considered. As

L211. 2018) concluded

L264. From figure
* * *

---

## Author Comment (AC1) · 1 Dec 2019

Dear referee,

The authors are very grateful for your careful reading and constructive comments. They are replied as below.

Best regards,
Bingchuan Nie

On behalf of all the authors.

**Technical corrections**
* * *
**Comment**: "*the four scenarios are not well designed and based on very weak hypothesis. There are several caveats in the design of the future scenarios (Sections 4.2 and 4.3). Tropical Cyclone Intensification (TCI) – this is taken into account by substituting the wind speed of the actual typhoon with the 50-year and 100-year return period wind speed. The reasoning between the choice of those return periods and the link with TCI should be better justified with supporting literature. Sea Level Rise (SLR) – the authors claim their approach in estimating sea level rise is advanced because they consider local changes in mean sea level, i.e. they extrapolate until 2050 the mean sea level from a tide gauge dataset from 1960 to present. There is an extensive literature on global and regional sea level rise projections that is overlooked in this paper. Taking the data from one tide gauge and projecting in the future with a polynomial fitting curve do not take into account that the rate of sea level rise will change in the future due to unprecedented external factors, i.e. global warming. An accurate estimate of SLR should consider the contribution to mean sea level from steric sea-level rise, dynamic sea-level change, glaciers and ice mass loss from glaciers, Greenland and Antarctic ice sheet contributions, as well as land-water storage changes and the glacial isostatic adjustment. There are regional projections available, see Jackson, Luke P., and Svetlana Jevrejeva. 'A probabilistic approach to 21st century regional sea-level projections using RCP and high-end scenarios.' Global and Planetary Change 146 (2016): 179-189.*"

**Reply**: The main focus of this work is on the potential inundation caused by storm tide. We want to provide the details of the potential inundation hazard of Southeast China coast considering TCI and SLR, which serves the complete risk assessment. Moreover, what we want to conveyed to the reader most is that TCI and SLR (considering their non-stationary and non-uniform effects) need more attention in the future risk analysis of storm tide inundation.

The influential factors of SLR are plenty and complex, just as the referee listed. To predict future SLR, two solutions can be usually found in literatures. One is based on the climate models such as Coupled Model Inter-comparison Project Phase 5 (CMIP5). The other one is purely data-driven prediction, i.e. estimating SLR based on statistically analyzing the long-term measurement of sea level. We chose the latter one because of simpler procedure. The comparison between the global averaged SLR reported in IPCC AR5 and our result are carried out. It shows that the SLR we obtained is reasonable, it falls in the situation between RCP 4.5 and RCP 6.0.

TCI and SLR are both related to global warming. Exploring their internal relationships is an interesting topic and must be full of challenges as well. This work is a case analysis of the potential storm tide inundation, in which extreme wind speed with specific return-period and SLR are regarded as two independent variables. However, it is acceptable, since return-period itself is measurement of possibility, and the study area is possible be harassed by typhoon with extreme wind speed of 100-year return period even at present.

According to the referee's comment, reference predicting the SLR via climate models is added in the manuscript, and more description is added for the design of the cases to show that two cases considering the impacts of TCI and SLR (case S3 and S4) are investigated based on the worst situation at present (S2).
* * *
**Comment** : "*one of the strengths of the paper would be to take advantage of the wave-current models coupling by showing waves results for the different scenarios, that are instead completely absent. The paper claim they are using a wave-hydro coupled model, but then the results are presented only in terms of water elevation and inundation maps, with no mention of waves. It would have been interesting to see for example the importance of the wavecurrents interactions, or at least maps of significant wave height under the different scenarios.* "

**Reply**: The tide-surge-wave coupling effect including the wave-current interaction for the same area of interest has already been analyzed in another paper of us (Wuxi, Q.Y., Li, J.C., and Nie, B.C.: Effects of tide-surge interaction and wave set-up/set-down on surge: case studies of tropical cyclones landing China's Zhe-Min coast, Theor. Appl. Mec. Lett., 8, 153-159, 2018.). It demonstrates that the coupling effects do have significant impact on the water elevation, which suggests coupling effects should be taken into account for hazard assessment of inundation. That motivates us to use the coupled model in this work.

Taking case S1 as an example, its wave fields and current are shown as Figure R1.
In this work, we focus on the storm tide inundation, which implies the water depth of the region we care most is shallow. Waves have contributed to water elevation and current mainly at surf zone, and became insignificant there. That's why those results are not presented in the manuscript.
———————————————————

**Comment**: "*the paper is poorly written; Section 3.1 is not needed. ADCIRC-SWAN are standard models already described in the literature and if the authors have not changed anything in the equations, there is no need to state them.*"
**Reply**: According to the referee's suggestion, the description of ADCIRC+SWAN model in section 3.1 is simplified. Only the most basic equation is remained to give a better explaining of the coupling manner among tide, surge and wave. It is also helpful for distinguishing of water elevation contributed by SLR or TCI, qualitatively.
———————————————————

**Comment**: "*the results are for just one typhoon and one specific location and it is complicated to get what is the advance in science presented in this paper or the impact of the results presented. The significance of the results, if any, is not conveyed to the reader. The results are just for one typhoon and for one SLR level and two enhanced wind speed scenarios. The paper does not present how these results can be generalized, for example focusing on the processes, or presenting a brand new methodology.*"

**Reply**: This work serves the risk analysis of storm tide inundation. Our results show that non-stationary TCI of 100 year return-period together with SLR equivalent to the situation of 2100s are able to double the potential inundation area of the worst situation at present. That may knoll alarm clock, climate change is able to deteriorate the storm tide inundation dramatically. It strongly suggests that climate change impact should be adopted in the future risk analysis of storm tide inundation, which is usually neglected at present. More specifically, TCI and SLR (the most direct two factors), which has significant temporally non-stationary and spatially non-uniform effects, must be involved for future risk analysis of storm tide inundation. That's the standpoint of this work.

To assess the local storm tide inundation hazard under climate change, a frame work is designed and implemented in this work. It integrates the trend analysis, numerical analysis and GIS-based analysis. The trend analysis takes the spatially non-uniform and temporally non-stationary effects of TCI and SLR into consideration, which is usually neglected at present. As for the numerical analysis, the surge-tide-wave coupling model is used to predict water elevation with higher precision. At last, relied on the GIS platform, storm tide inundation can be provided including the details such as inundation area, regions and duration, which is closely related to risk analysis. The frame work is quite satisfactory and has high feasibility, and it can be used for assessing storm tide inundation hazard for other regions.

Moreover, the study area in this work, Southeast China coast, is a very typical storm tide prone area over China and worldwide just like the north coast Gulf of Mexico. Typhoons make landfall there tends to have very strong intensity and cause extensive economic losses and shocking live losses. For example, about 70 people dead or missing when typhoon Lekima (201909) made land fall just at the study area about three months ago. The literature review shows that inundation caused by storm tide at Southeast China coast is scarcely studied, let alone the long-term hazard assessment of storm tide inundation considering potential TCI and SLR. We examined the inundation considering TCI and SLR effects based on the severest situation there

that typhoon with largest wind speed makes landfall perpendicular to the coastline during astronomical high tide. The corresponding results will provide reference for the prevention and mitigation of storm tide inundation hazard and future coastal management there.

**Specific Comments**

———————————————————————

**Comment**: "*L2. 'trend analysis, numerical analysis and GIS-based analysis'– authors should be more specific*"
**Reply**: More description is added. "It integrates the statistical trend analysis considering temporally non-stationary and spatially non-uniform effects, numerical analysis taking into account the tide-surge-wave coupling effect and GIS-based analysis for inundation evaluation."

———————————————————————

**Comment**: "*L4. 'non-stationary'– it is not clear how the authors take into account the non-stationarity of tropical cyclones and mean sea level.* "
**Reply**: More description is added. "In the trend analysis, the potential TCI and SLR are estimated by analyzing the long-term historical Tropical Cyclone (TC) data relied on the non-stationary extreme value theory and Mean Sea Level (MSL) data via extrapolation"

———————————————————————

**Comment**: "*L9. Maximum water elevations should be compared with values during present conditions, to understand the magnitude of changes.* "
**Reply**: The comparison is added. "The maximal water elevations of the worst situation at present without considering TCI and SLR (i.e. case S2) are 6.06 m, 5.82 m and 5.67 m around Aojiang, Feiyunjiang and Oujiang river estuaries, respectively. Whereas, the maximal water elevations for the three estuaries would increase to 7.02 m, 6.67 m and 6.44 m, respectively, when the non-stationary extreme wind speed of 100-year

recurrence period and SLR equivalent to the situation of 2100s (i.e. case S4) are taken into account. The potential inundation area of case S4 would expand by 108% to about 798 km$^2$ compared with case S2."

————————————————

**Comment**: "*L22. It is worth mentioning typhoons impacting the Chinese coast, as Haynan killed 6300 people, but in the Philippines.* "
**Reply**: It has been revised as "..., TC Haiyan in 2013 resulted in 6300 dead and 1061 missing in Philippines alone (Lagmay et al., 2015).".

————————————————

**Comment**: "*L23-33. This literature review does not hold together. Risk assessment, vulnerability and hazard assessment are randomly mentioned, but the previous studies are not presented in a coherent way, it is not clear what the present work is adding to previous literature.* "
**Reply**: This paragraph reviews the works related to risk assessment, consists of hazard and vulnerability assessments, of storm tide inundation. It is rewrote in the revised manuscript.
In addition, more description of the previous work and the contribution of this work are added, say, "In general, the impacts of TCI and SLR are scarcely considered at present with focus on the risk assessment of storm tide inundation. In a few relevant works available, TCI or SLR are usually artificial gave or obtained according to the existing global averaged results, which means the spatial inhomogeneous and non-stationary effects are neglected. Moreover, considering the significant regional differences, impacts of TCI and SLR on the storm tide inundation at typical storm surge prone areas remain to be carried out as well.".

————————————————

**Comment**: "*L24. 'empirical model' – authors should be more specific*"
**Reply**: More detail is added. "Based on the surge response function, which gives the relations between maximum surge elevations and hurricane parameters, Hus et al. (2018) examined the flood risk of northern Gulf of Mexico coast exposed to storm

surge."

———————————————————

**Comment**: "*L28. 'surrogate model' – authors should be more specific*"
**Reply**: More detail is added. "Taflanidis et al. (2013) carried out the risk estimation of TC waves, water elevations, and run-up for TC passing by the Island of Oahu. In their work, a response surface methodology fed by information from precomputed database is adopted to evaluate the maximum wave height and water level according to the hurricane parameters. "

———————————————————

**Comment**: "*L32. 'the precisions of water elevation prediction models in the afore-mentioned works are limited resulting in quite coarse hazard assessment' – this should be supported by evidence. Hazard assessment usually means to evaluate the probability/evidence of an event. Not clear what is a "coarse" hazard assessment. With a coarse resolution model?* "
**Reply**: The analysis of the previous hazard assessments are more specified: "In general, the hazard assessments in the aforementioned works remain to be improved: 1) the precisions of models for water elevation predicting are unsatisfactory, and 2) the intensity of storm tide inundation including area, depth and duration needs to be com-prehensively quantified". More specifically, water elevation prediction models of the present hazard assessments did not considering the tide-surge-wave coupling effect, and the coupling effect has significant influence on water elevation and subsequent inundation intensity. That is supported by the paragraph following it.
Hazard assessment is aimed to evaluate the natural attributes of an event, i.e. the intensity (area, depth and duration) and frequency of storm tide inundation in this work. For a specific region, the frequency of storm tide inundation hazard is related to the frequency of TC passing by there quantified by the return-period.

———————————————————

**Comment**: "*L36. 'continual renovation' – it does not sounds as the right term*"
**Reply**: The sentence is rewrote as "The ADCIRC model developed by Luettich and his

colleagues (Luettich et al., 1992), which has been improved continuously, are widely applied in the academic and engineering communities".

———————————————————

**Comment**: "*L44/45 'They found : : : 28'* "
**Reply**: We don't get the referee's intention. We rewrite the sentence as "They found that the relative error of water elevation due to tide-surge coupling effect can be as high as 28% maximal.".

———————————————————

**Comment**: "*L54. Cite Walsh, Kevin JE and McBride, John L and Klotzbach, Philip J and Balachandran, Sethurathinam and Camargo, Suzana J and Holland, Greg and Knutson, Thomas R and Kossin, James P and Lee, Tsz-cheung and Sobel, Adam and others. (2016). Tropical cyclones and climate change. Wiley Interdisciplinar Reviews: Climate Change, 7 (1), 65-89 and references therein.* "
**Reply**: This article has been already cited in the original manuscript.

———————————————————

**Comment**: "*L54. 'NWP' - missing the acronym definition*"
**Reply**: It has been revised as "Recently, Wang et al (2016) examined the extreme wind speeds in northwest Pacific Ocean (NWP) and South China Sea (SCS)".

———————————————————

**Comment**: "*L60/61 'increasing rate: : : level': this sentence is not clear, in 0.06 m units are not for a rate.* "
**Reply**: It is "0.06 m per year", the referee missed the words "per year".

———————————————————

**Comment**: "*L67. What is the global TCI? How does it compare with the regional one the authors are using?* "
**Reply**: The global TCI means that tropical cyclone intensification is estimated based on the global typhoon database. For example, Elsner et al. (2008) reported that significant wind speed upward trends of 0.3-0.09 $ms^{-1} \cdot yr^{-1}$ can be observed for the strongest TCs. As for the regional TCI, we used the non-stationary extreme value

theory (Wang, L.Z. and Li, J.C.: Estimation of extreme wind speed in SCS and NWP by a non-stationary model, Theor. Appl. Mec. Lett., 6, 131-138, 2016.) to obtain the local extreme wind speed with different return periods. The extreme wind speed is closely related to the historical typhoon database, and the spatial difference of typhoon worldwide is huge. Thus, we did not compare the regional extreme wind speed with the global ones.

————————————————

**Comment**: "*L73 '1.73 billion USD' – reference needed*"
**Reply**: This result is based on the statistical analysis of the annual report of Chinese marine hazard issued by Ministry of Natural Resources of P.R. China (see Figure R2). The data resource is provided in the revised manuscript along with the website assessing to the reports provided in the section code and data availability.

————————————————

**Comment**: "*L159. What is the length scale? From where Vm, U0, RG and Rm are taken?*"
**Reply**: The length scale RG is specified as "the length scale of environmental process (of order of 500 km)". Vm, U0 and Rm are from China Meteorological Administration, we have stated that in the section, code and data availability, already.

————————————————

**Comment**: "*L163. From where Pc and Pn and track of TC are taken?*"
**Reply**: They are from China Meteorological Administration, we have stated that in the section, code and data availability, already.

————————————————

**Comment**: "*L192-193. The AR5 IPCC sea level rise projections are given as the mean sea level 2081–2100 average relative to 1986–2005 average. While the authors are comparing 2006 with 2100. How is it imposed the SLR in the model?*"
**Reply**: Using our results, mean sea level during 2081-2100 is 7456.2 mm, which is 477 mm higher than mean sea level during 1981-2005. Comparing with that reported in AR5 are modified. "The historical data in Kanmen gauge shows that MSL during

1986-2005 is 6979 mm, while the extrapolation demonstrates that the local MSL during 2081-2100 would be 477 mm higher relative to 1986-2005. That is to say the SLR we obtained is between the situations of RCP 4.5 and RCP 6.0. It should be noted that since the historical data only reflects the response of climate system to influential factors in the past, the real SLR may speed up or slowdown dependent on the factors such as greenhouse gas emission in the future."

––––––––––––––––––––––––––––

**Comment**: "*Table 1. Scenarios S3 and S4 are not well designed, as it is impossible to understand if the changes are due to MSL changes or wind changes.* "

**Reply**: S3 and S4 are two potential cases based on the severest situation that typhoon with largest wind speed makes landfall perpendicular to the coastline during astronomical high tide. Examining these two cases, we want to conveyed to the reader is that climate change (SLR and TCI, mainly) can deteriorate the inundation hazard significantly, and it should be considered in the future risk analysis.

As for the water elevation due to MSL and TCI alone, they were just planned to be distinguished qualitatively. "Since the period of astronomical tide cycle is a few times larger than the duration of TC passing by for a specific location, the increase of tide level can be regarded as the quasi-steady process of increasing the water depth. That is similar to SLR, but much larger amplitude. It implies that the increasement of water elevation contributed by SLR can be estimated roughly by the tide-surge coupling effect. Taking S4 for example. The MSL of S4 is 0.51 m higher than that of S2. That means increase of water elevation caused by SLR will be a little bit less than 0.51 m, because less wave induced surge occurs for higher water depth. While, the total increase of water elevation by SLR and TCI, 0.80 m, suggests that increase of water elevation caused by TCI could be larger than 0.29m. The qualitative results can be obtained for S3 similarly, i.e. TCI could cause more than 0.16 m of water elevation increasement, while SLR can caused water elevation increasement a little bit less than 0.19 m. In all, both TCI and SLR are important factors should be involved in the future long-term hazard assessment of storm tide."

Actually, we did simulated the cases considering MSL and TCI effects independently (cases C1 and C2) and simultaneously (case C3) to carry out the comparison analysis of their contributions. The parameters and main results of those cases are shown as in Table R1. The water elevations for those three cases are presented in Figure R3 as well. The maximal water elevation without considering TCI and SLR, i.e. S2 in the manuscript, is 5.9 m. Comparing C1 with S2, it shows that the maximal increment of water elevation due to SLR effect is 0.26 m. While, sea level of C1 is 0.32 m higher than S1. That is to say the maximal increment of water elevation due to SLR effect is less than the SLR itself, and increment of water elevation due to SLR is less than SLR only a little bit for most part of the computational domain. Comparing C3 and C2, the same conclusion can be obtained. The main reason is that higher water elevation will cause less wave breaking and wave induced surge. Those quantitative results support the qualitative analysis above.

Since qualitative analysis can already give the conclusion that both TCI and SLR are important factors should be involved in the future risk assessment of storm tide inundation, those quantitative results are not presented in the manuscript.
* * *
**Comment**: "*L201. Is the maximum wind kept constant during the simulation? Or how the wind of the real typhoon is adjusted to reach the 50- and 100-year return period?* "
**Reply**: The maximal wind speed of the assumed typhoon varies with time. In each time step, the amplification coefficient between the assumed typhoon and the strongest typhoon are the same. The procedures of reconstructing wind field are added. "Based on the parameters in table 1, the variation of wind fields can be reconstructed to consider the landing moment and TCI effects. More specifically, the history of maximal wind speed is evaluated by multiplying the actual wind speed of Saomai by the amplification coefficient firstly. Where, the amplification coefficient is determined by the maximal wind speed of assumed TCs and that of Saomai focusing on the landing moment. Then, the wind fields with interval of an hour are reconstructed based on the TC wind field model. Adjusting the time coordinate of the assumed TC

base on Saomai make it lands during astronomical high tide, thus, the location of the TC center can be determined. As for MSL, it can be involved by increasing the water depth of the computational domain. Taking astronomical tide, wind and pressure fields as input, water elevation can be obtained relied on the aforementioned hydrodynamic coupled model."

————————————————

**Comment**: "*L210. Reference needed.* "
**Reply**: It is based on the annual report of national marine hazard of 2006. Data resource is added.

————————————————

**Comment**: "*L212-219. Poorly written. Not clear what "coupling effect" means and how it affects the surge (shown in Figure 6). In addition, what are the processes behind the "nonlinear tide-surge effect"? High water depth (due to high tide) means reduced bottom friction and possibly increased surge. At the same time, reduced wind stress (inversely proportional to the water depth) can lead to a decrease in the surge. So, it looks like in this case the wind could be more important than the bottom friction effect since the surge reduces. The authors should explain better the effects of the different processes, including waves.* "
**Reply**: The "coupling effect" here means the nonlinear coupling effects among surge, tide and wave. If the coupling effects are not planned to be considered, the calculations of tide, surge and wave can be ran separately. Then, their contributions to the water elevation can be superposed to predict real water elevation. However, the real water elevation is unequal to the superposed results because of the nonlinear coupling effects. The water elevation due to coupling effects can be quantified by the difference between the real water elevation and superposed results. The relative errors of water elevation due to coupling effects were provided and discussed quantitatively in another paper of us (Wuxi, Q.Y., Li, J.C., and Nie, B.C.: Effects of tide-surge interaction and wave set-up/set-down on surge: case studies of tropical cyclones landing China's Zhe-Min coast, Theor. Appl. Mec. Lett., 8, 153-159, 2018.).

The main physical processes of the coupling effects include: 1) wave can contribute to water elevation via wave breaking; 2) water elevation can affect wave breaking processes in turn through change the wave breaking limit, bottom friction and so on; 3) current is coupled with water elevation which is controlled by the continuity equation, and the wave-current exists as well.

This paragraph is rewrote in the revised manuscript. "The difference of water elevation between S2 and S1 linearly superposed on the tide level difference (about 4.1 m) is presented in figure 6. It shows that water elevation of S2 can be 1.2 m (maximal) less than that of S1 linearly superposed on the tide level difference, which occurs at area north of the TC track and in the bays. While, that at most part of the computational domain is slightly larger than zero. In other words, the actual increasement of water elevation due to higher tide level is less than the increasement of tide level itself, and the their relative difference can be 29% maximal. The nonlinear coupling effect between tide and surge through wave are responsible for that. More specifically, waves with larger height can survival in higher water depth, which implies wave induced surge becomes less when the astronomical tide level increases. As for the temporally evolution of water elevation difference caused by the coupling effects, Wuxi et al. (2018) concluded that it experiences insignificant premonitory fluctuation, remarkable rise/drop and residual oscillating with a gentle decay of magnitude."

————————————————————

**Comment**: *"L230. Again authors should explain why it is smaller/larger."*
**Reply**: SLR effect is quite similar to the situation that typhoon lands during different tide levels, since the time scale of the period of astronomical tide is a few times larger than the duration of typhoon passing by a specific location. Thus, water elevation due to SLR effect can be explained and estimated roughly according to the nonlinear tide-surge coupling effect.

This paragraph is rewrote and more description is added in the revised manuscript. "Since the period of astronomical tide cycle is a few times larger than the duration of TC passing by for a specific location, the increase of tide level can be regarded as the

quasi-steady process of increasing the water depth. That is similar to SLR, but much larger amplitude. It implies that the incresement of water elevation contributed by SLR can be estimated roughly by the tide-surge coupling effect. Taking S4 for example. The MSL of S4 is 0.51 m higher than that of S2. That means increase of water elevation caused by SLR will be a little bit less than 0.51 m, because less wave induced surge occurs for higher water depth. While, the total increase of water elevation by SLR and TCI, 0.80 m, suggests that increase of water elevation caused by TCI could be larger than 0.29m. The qualitative results can be obtained for S3 similarly, i.e. TCI could cause more than 0.16 m of water elevation increasement, while SLR can caused water elevation increasement a little bit less than 0.19 m. In all, both TCI and SLR are important factors should be involved in the future long-term hazard assessment of storm tide."

—————————————————————

**Comment**: "*Figure 5. The authors should use the same limits for the colorbar to help the visual comparison of the different scenarios. Figure 5 for scenario S3 and S4 is not described in the text.*"

**Reply**: The main message expected to convey to the reader through figure 5 is the details of the surge at north of the landing location, which is directly related to the severe inundation (the focus of this work). The maximal water of (a) is much lower than the other three cases. If the same up limit is used for all subfigures, it will be hard to quantify the maximal water elevation of interest, and the details of surge at south of the landing location will become unclear as well for (a). On the other hand, we prefer the same lower limits for all subfigures. Otherwise, readers may mistake the regions in blue color in (b)-(d) as negative surge at first glance. Therefore, -3.5 m (lower limit of (a)) is treated as the lower limits, and the up limit of each subfigure is determined by its maximal water elevation. Using those limits, figure 5 is replotted.

—————————————————————

**Comment**: "*L227-237. It is very difficult to understand which are the processes that are into play. Also because in scenario S3 and S4 both the wind and MSL are*

*changing.* "
**Reply**: This paragraph is rewrote in the revised manuscript as mentioned above.
———————————————————

**Comment**: "*Figure 7. The two panels can be merged in one figure to compare better the different scenarios.* "
**Reply**: Subfigures 7 (a) and (b) are merged into one figure.
———————————————————

**Comment**: "*Figure 8/9. Is the red line the actual coastline? Is the blue area where wetting and drying is allowed in the model? The authors should mark the landing location of the typhoon too.* "
**Reply**: The red lines are the boundaries of the inundation regions. More description is added in the caption of figures 8 and 9, in which the track and landing location of typhoon are added. The wetting and drying function is on during calculation.

**Technical corrections**

———————————————————

**Comment**: "*L26. (2017) analyzed*"
**Reply**: The space character is added between "(2017)" and "analyzed".
———————————————————

**Comment**: "*L41. 2011) and*"
**Reply**: The space character is added between "…2011)" and "and".
———————————————————

**Comment**: "*L49. 2005) examined*"
**Reply**: The space character is added between "…2005)" and "examined".
———————————————————

**Comment**: "*L54. considered. As* "
**Reply**: The space character is added between "considered." and "As".
———————————————————

**Comment**: "*L211. 2018) concluded*"
**Reply**: The space character is added between "...2008)" and "concluded".

————————————————————

**Comment**: "*L264. From figure*"
**Reply**: The figure number "11" is in the next page in the original manuscript, the referee missed it.

————————————————————

[Figure]

[Figure]

**Fig. 1.** Figure R1: Wave fields and current for S1. (a) and (b) are significant wave height (m) at two moments; (c) is the current. The red line and dots are the track and center of typhoon, respectively.

[Figure]

**Fig. 2.** Figure R2: Direct economic loss in China due to storm tide. The red line indicates the annual averaged direct economic loss is about 12.1 billion RBM (1.732 billion USD). Data resource: http://www.nmd

**Fig. 3.** Figure R3: Water elevation (m) at the land moment. (a)-(c) are for case C1-C3, respectively. The parameters and maximal water elevation for those cases are given in Table R1.

Table R1. Cases considering MSL and TCI independently/together. S2 is one of the scenarios examined in the manuscript. C1-C3 are scenarios we simulated but didn't appear in the manuscript.

| Cases | MSL (mm) | TC intensity (m/s) | Landing moment | TC track | Maximal water elevation (m) |
|-------|----------|--------------------|----------------|----------|------------------------------|
| S2 | 7013.6 | 60.0 | High tide | Saomai | 5.90 |
| C1 | 7333.6 | 60.0 | High tide | Saomai | 6.16 |
| C2 | 7013.6 | 89.4 | High tide | Saomai | 7.32 |
| C3 | 7333.6 | 89.4 | High tide | Saomai | 7.63 |

**Fig. 4.** Table R1

---

## Referee Comment (RC2) · Anonymous Referee #2 · 27 Dec 2019

Manuscript: **Assessing potential storm tide inundation hazard under climate change: a case study of Southeast China coast** by Bingchuan Nie et al

NHESS – 2019 -284

December 2019

Authors present a methodology for delineating storm induced inundation including the role of Tropical Cyclone Intensification and Sea-Level-Rise. The methodology is then applied to assess the effects of TCI and SLR on storm-induced inundation in the SE China coast. The topic is in line with one of the main targets of NHESS and, in this sense, it could be of interest for NHESS's readers. However, the manuscript present some points that need further development before being published.

In what follows, some observations/comments/suggestions are given.

**[C-1]** According to the authors, the main aim of the manuscript is to present a methodology to account for the role of TCI and SLR on storm-induced inundation. This methodology will also consider the role of waves on inundation. After reading the manuscript is not clear which is the originality or innovation of the work. Authors apply a standard approach where they use a widely employed hydrodynamic model suite and just modify forcing/boundary conditions. Then, they apply the methodology to a specific site for 1 reference scenario + 3 modified ones. Thus, in reality, the main contribution of the paper seems to be assessing the changes in storm-induced inundation in a specific site under different scenarios. However, if this is the real objective of the paper, selected scenarios need to be better defined/selected or justified and the analysis must be deeper covered.

**[C-2] Methodology**

This section must be improved. Although the main objective is to present and apply the methodology, the half of the section (from lines 80 to 90) is not giving any details about methodology but providing some general text. The rest of the text is just giving a brief outline overview on some used tools/models. At its present form, this section can be fully removed without affecting the manuscript. The best option should be rewriting this section by putting emphasis on describing the general methodological framework (e.g. to describe steps in figure 1) and how to apply it. For instance, authors select as a base case scenario conditions recorded during a TC and then, they propose some scenarios. It is VERY important to properly describe how to build future scenarios to cope with time variation in TCI and SLR. At present this info is partially (insufficiently) covered in section 4, but needs to be included here. All these steps need to be well justified and well described and, since this is an IMPORTANT part of the analysis (scenario selection), this section is the best place to describe how to do it.

**[C-3] Hydrodynamic surge-tide-wave coupled model**

This section is superfluous. Since authors do not modify the model and the model is widely used and very well-known and referenced, it would be enough to mention it with proper referencing. The most important part is how to select conditions to be simulated and, details on model setting (grid, boundary conditions, etc.). All this info could be integrated within section 2 (Methodology).

**[C-4] Case study**

**4.2** The first part of this section needs to be better described and included in methodology (section 2). You mention that you have segmented time series in 50-years long time series which are fitted to an extreme distribution (Weibull). Why Weibull? Is this the best distribution? How can you justify it ($r^2$-values)? Once you have fitted all time series, what to do next (step 3)? Are you doing trend-analysis on fitted parameters to see time evolution of distribution's parameters? If yes, please be explicit. The text does not clearly describe this step.

**4.3** Why did you select a specific TC to do the analysis. Conditions for this TC does not seems to be really strong since recorded wind speed (60 m/s) are weaker than the wind speed associated with a return period of 20 year (figure 3). Please comment about this.

The selection of SLR scenario is just an extrapolation of recorded local conditions. This is a very simple approach and it need to be better justified. It has to be considered that using this approach is not accounting for any possible acceleration in SLR and, in this sense, it has not too much meaning to compare with IPCC scenarios as authors do in lines 190-193. In any case the ideal situation will be to add to local SLR the expected changes due to CC which would result in rates larger than the used by authors.

Considering the previous comments, scenarios used by authors need to be reformulated or much better justified. It should be great if authors dedicate a larger effort to this task. They need to consider that since no significant novelty in methodology is provided, the best contribution they can do is to perform a solid assessment. Otherwise it would be an academic exercise without too much practical interest.

**4.4** Results presented in Fig 5 could be much better compared if you use the same scale for all figures. Also, which is the relevance of representing water level at the sea, especially when you are also plotting the component associated to astronomical tide? If you mention that one of the advantages of your approach is accounting for the wave contribution, why do not show wave heights? They will be modulated by water level and, thus, you can assess how the hazard component associated to waves does change from one scenario to other one.

It has not too much meaning to compare different scenarios at different tide conditions unless you want to specifically assess the role of the astronomical tide. If you want to assess the contribution of TCI and SLR you just need to concentrate in compare any scenario under the same tide condition. Please, simplify.

Are water levels represented in Figure 7 also including the wave contribution (run up)? If so, which is the difference in this contribution between scenarios? Thus, you can account which is the contribution of each component (TCI, SLR, waves to differences in total water level)? Why don't you include all graphs within a single figure (it should be the best way to compare them)?

**4.5** Results showed in this section are only relevant if tested scenarios are relevant/representative.

**[C-5] Conclusions**

This section needs to be modified after implementing previous recommended changes.

Figure 10 is not needed.

---

## Author Comment (AC2) · 13 Jan 2020

Dear referee,

Thank you for your comments, they are very helpful for improving our paper. The replies and corresponding revisions are described as below.

Best regards,

Bingchuan Nie

On behalf of all the authors.

[Figure]

**Comments and Replies**
* * *
**Comment:** *According to the authors, the main aim of the manuscript is to present a methodology to account for the role of TCI and SLR on storm-induced inundation. This methodology will also consider the role of waves on inundation. After reading the manuscript is not clear which is the originality or innovation of the work. Authors apply a standard approach where they use a widely employed hydrodynamic model suite and just modify forcing/boundary conditions. Then, they apply the methodology to a specific site for 1 reference scenario + 3 modified ones. Thus, in reality, the main contribution of the paper seems to be assessing the changes in storm-induced inundation in a specific site under different scenarios. However, if this is the real objective of the paper, selected scenarios need to be better defined/selected or justified and the analysis must be deeper covered.*

**Reply:** The frame work which integrating the trend analysis, numerical analysis and GIS-based analysis is new. It seems quite satisfactory and feasible when assessing the storm tide inundation hazard of the study area of interest. It provide a choice for storm tide inundation hazard assessment at other regions.

Moreover, this work is expected to be helpful for promoting the hazard assessment of storm tide inundation, since the non-uniform and non-stationary TCI and SLR impact is usually neglected at present, and the wave influence does not draw enough attention neither when assessing the storm tide INUNDATION. Whereas, our results shows that they are able to deteriorate the storm tide inundation dramatically, and suggests that they should be adopted in the future risk analysis of storm tide inundation.

On the other hand, the study area Southeast China coast considered in this work is a very typical storm tide prone area over China and worldwide just like the north coast Gulf of Mexico. For example, about 70 people dead or missing when typhoon Lekima (201909) made land fall just at the study area about a few months ago. The literature review shows that inundation caused by storm tide at Southeast China coast is scarcely studied, let alone the long-term hazard assessment of storm tide inundation considering potential TCI and SLR. Those typical cases considered can provide reference for the prevention and mitigation of storm tide inundation hazard and future coastal management there.

The introduction section is strengthened to present the innovation more clearly.

———————————————————

**Comment:** *[Methodology] This section must be improved. Although the main objective is to present and apply the methodology, the half of the section (from lines 80 to 90) is not giving any details about methodology but providing some general text. The rest of the text is just giving a brief outline overview on some used tools/models. At its present form, this section can be fully removed without affecting the manuscript. The best option should be rewriting this section by putting emphasis on describing the general methodological framework (e.g. to describe steps in figure 1) and how to apply it. For instance, authors select as a base case scenario conditions recorded during a TC and then, they propose some scenarios. It is VERY important to properly describe how to build future scenarios to cope with time variation in TCI and SLR. At present this info is partially (insufficiently) covered in section 4, but needs to be included here. All these steps need to be well justified and well described and, since this is an IMPORTANT part of the analysis (scenario selection), this section is the best place to describe how to do it.*

**Reply:** According to your suggestion, this section is rewritten in the revised manuscript. More details about the non-stationary TCI model including the basic equations for estimating the maximal wind speed of TC under climate change is added. And, the reconstruction of potential TC's wind field at each time step from the maximal wind speed and TC track is added. Thus, how to build the wind fields of future scenarios considering TCI and landing moment is fulfilled. On the other hand, the hydrodynamic surge-tide-wave coupled model in section 3 is integrated within the section of methodology.
* * *
**Comment:** *[Hydrodynamic surge-tide-wave coupled model] This section is superfluous. Since authors do not modify the model and the model is widely used and very well-known and referenced, it would be enough to mention it with proper referencing. The most important part is how to select conditions to be simulated and, details on model setting (grid, boundary conditions, etc.). All this info could be integrated within section 2 (Methodology).*

**Reply:** Yes. The description of Hydrodynamic surge-tide-wave coupled model is simplified along with the description of the grid and boundary conditions details added. And these revisions are integrated within Section 2 in the revised manuscript as the referee suggested.
* * *
**Comment:** *[Case study] The first part of this section needs to be better described and included in methodology (section 2). You mention that you have segmented time series in 50-years long time series which are fitted to an extreme distribution (Weibull). Why Weibull? Is this the best distribution? How can you justify it (r2-values)? Once you have fitted all time series, what to do next (step 3)? Are you doing trend-analysis on fitted parameters to see time evolution of distribution's parameters? If yes, please be explicit. The text does not clearly describe this step.*

**Reply:** In the revised manuscript, the procedures for constructing the wind field is added according to reviewer's suggestion. The main details are as below. Once the extreme wind speed of different return period is obtained, the maximal wind speed at each time step is ready to be evaluated by multiplying the actual wind speed of Saomai by the amplification coefficient. Where, the amplification coefficient is determined by the maximal wind speed of assumed TCs and that of Saomai focusing on the landing moment. Then, the wind fields at each time step can be reconstructed based on the

[Figure]

TC wind field model. Further, adjust the time coordinate of the assumed TC to make sure the TC lands during astronomical high tide, thus, the track of the TC center can be determined. As for the non-stationary TCI model, it has been described in a previous paper of ours (Wang, L.Z. and Li, J.C.: Estimation of extreme wind speed in SCS and NWP by a non-stationary model, Theor. Appl. Mec. Lett., 6, 131-138, 2016), in which the Poisson distribution for the TC annual frequency and Weibull distribution for wind speed are justified carefully.

———————————————————

**Comment:** *[Case study] Why did you select a specific TC to do the analysis. Conditions for this TC does not seems to be really strong since recorded wind speed (60 m/s) are weaker than the wind speed associated with a return period of 20 year (figure 3). Please comment about this.*

**Reply:** Examining the historical typhoons passing by the study area from 1945, it shows that Saomai has the largest wind speed when LANDFALL. Moreover, Saomai struck the coast almost vertically, the most dangerous landing angle. In reality, Saomai did claimed 480 deaths and 2.5 billion USD economic loss according to the annual report of national marine hazard of 2006. That is to say Saomai is most typical TC landing the area of interest, thus, it was chose as the basic TC. In addition, for Saomai, we have observation data available for verification (Wuxi, Q.Y., Li, J.C., and Nie, B.C.: Effects of tide-surge interaction and wave set-up/set-down on surge: case studies of tropical cyclones landing China's Zhe-Min coast, Theor. Appl. Mec. Lett., 8, 153-159, 2018.). Those explanations are added in the revised manuscript as the reviewer suggested.

———————————————————

**Comment:** *[Case study] The selection of SLR scenario is just an extrapolation of recorded local conditions. This is a very simple approach and it need to be better justified. It has to be considered that using this approach is not accounting for any possible acceleration in SLR and, in this sense, it has not too much meaning to compare*

*with IPCC scenarios as authors do in lines 190-193. In any case the ideal situation will be to add to local SLR the expected changes due to CC which would result in rates larger than the used by authors.*

**Reply:** To predict future SLR, two solutions can be usually found in literatures. One is based on the climate models such as CMIP5 and upcoming CMIP6. The other one is purely data-driven prediction, i.e. estimating SLR based on statistically analyzing the long-term measurement of sea level. We chose the latter one because of simpler procedure. In fact, extrapolation of recorded local mean sea level do account for the possible acceleration in SLR, since it is a nonlinear extrapolation.

More details about the comparison between the results by extrapolation and that reported in AR5 are added in the revised manuscript. The AR5 of IPCC reported that the global SLR for 2081-2100 relative to 1986-2005 will likely be 260 to 550 mm for RCP 2.6, 320 to 630 mm for RCP 4.5, 330 to 630 mm for RCP 6.0, and 450 to 820 mm for RCP 8.5. The historical data in Kanmen gauge shows that MSL during 1986-2005 is 6979 mm, while the extrapolation demonstrates that the local MSL during 2081-2100 would be 477 mm higher relative to 1986-2005. That is to say the SLR we obtained is between the situations of RCP 4.5 and RCP 6.0. It should be noted that since the historical data only reflects the response of climate system to influential factors in the past, the real SLR may speed up or slowdown dependent on the factors such as greenhouse gas emission in the future.

――――――――――――――――――――――――――

**Comment:** *[Case study] Considering the previous comments, scenarios used by authors need to be reformulated or much better justified. It should be great if authors dedicate a larger effort to this task. They need to consider that since no significant novelty in methodology is provided, the best contribution they can do is to perform a solid assessment. Otherwise it would be an academic exercise without too much practical interest.*

**Reply:** Justification of the scenarios are added in the revised manuscript. The main points are: 1) the area of interest is very typical over China and worldwide just like the north coast Gulf of Mexico; 2) The basic TC Saomai is typical, since it has the largest wind speed during made landfall at the area of interest among the TC during 1945-2013 and struck the coast with the most dangerous landing angle; 3) scenario S2 in which typhoon has the same extreme of Saomai and makes landfall during the astronomical high tide is the worst situation could occur at present; 4) S3 and S4 are two typical future scenarios considering TCI and SLR based on S2.
* * *
**Comment:** *[Case study] Results presented in Fig 5 could be much better compared if you use the same scale for all figures. Also, which is the relevance of representing water level at the sea, especially when you are also plotting the component associated to astronomical tide? If you mention that one of the advantages of your approach is accounting for the wave contribution, why do not show wave heights? They will be modulated by water level and, thus, you can assess how the hazard component associated to waves does change from one scenario to other one.*

**Reply:** Yes, we have modified the scales of figure 5 with its subfigures updated. The water elevation is measured from the mean sea level, which is contributed by tide and surge directly and by wave indirectly. The water depth of the storm tide inundation region we care most in this work is so shallow that large waves is hard to survival there. Waves contributed to water elevation and current is obvious mainly in surf zone. The evolution of components of water elevation due to tide, wave and surge and their coupling effect have been already analyzed quantitatively in previous paper of us (Wuxi, Q.Y., Li, J.C., and Nie, B.C.: Effects of tide-surge interaction and wave set-up/set-down on surge: case studies of tropical cyclones landing China's Zhe-Min coast, Theor. Appl. Mec. Lett., 8, 153-159, 2018.). In addition, the spatial distribution of wave impact can also be figured out from figure 6. As for the wave heights and the main physical processes of the tide-surge-wave coupling, one can refer to the second and 26th reply

to referee 1, respectively.

——————————————————————

**Comment:** *[Case study] It has not too much meaning to compare different scenarios at different tide conditions unless you want to specifically assess the role of the astronomical tide. If you want to assess the contribution of TCI and SLR you just need to concentrate in compare any scenario under the same tide condition. Please, simplify.*

**Reply:** According to the referee's suggestion, the comparison between S1 and S2 which have different tide condition is simplified. And more discussion about the contributions of TCI and SLR to water elevation are added, see the reply to the next comment.

——————————————————————

**Comment:** *[Case study] Are water levels represented in Figure 7 also including the wave contribution (run up)? If so, which is the difference in this contribution between scenarios? Thus, you can account which is the contribution of each component (TCI, SLR, waves to differences in total water level)? Why don't you include all graphs within a single figure (it should be the best way to compare them)?*

**Reply:** Your comment is very helpful. Subfigures 7 (a) and (b) are merged into one figure in the revised manuscript. As for wave, the contribution to water elevation at surf zone via wave breaking is much more significant than the run up for storm tide inundation. The former can increase the water elevation more than 10 percent as surf zone (Wuxi, Q.Y., Li, J.C., and Nie, B.C.: Effects of tide-surge interaction and wave set-up/set-down on surge: case studies of tropical cyclones landing China's Zhe-Min coast, Theor. Appl. Mec. Lett., 8, 153-159, 2018.).

The qualitative discussion on the contributions due to TCI and SLR, are added in the revised manuscript as below. Since the period of astronomical tide cycle is a few times larger than the duration of TC passing by for a specific location, the increase of tide level can be regarded as the quasi-steady process of increasing the water depth.

That is similar to SLR, but much larger amplitude. It implies that the increasement of water elevation contributed by SLR can be estimated roughly by the tide-surge coupling effect. Taking S4 for example. The MSL of S4 is 0.51 m higher than that of S2. That means increase of water elevation caused by SLR will be a little bit less than 0.51 m, because less wave induced surge occurs for higher water depth. While, the total increase of water elevation by SLR and TCI, 0.80 m, suggests that increase of water elevation caused by TCI could be larger than 0.29m. The qualitative results can be obtained for S3 similarly, i.e. TCI could cause more than 0.16 m of water elevation increasement, while SLR can caused water elevation increasement a little bit less than 0.19 m. In all, both TCI and SLR are important factors should be involved in the future long-term hazard assessment of storm tide. To prove that, results of a few new cases considering MSL and TCI effects independently has been presented in Figure R3 of the reply letter to referee 1.

––––––––––––––––––––––––––––––––––

**Comment:** *[Case study] Results showed in this section are only relevant if tested scenarios are relevant/representative.*

**Reply:** As described above, the area of interest is a very typical storm tide prone area over China and worldwide from the view of intensity and occurrence frequency of TC and the damages caused. On the other hand, the four scenarios are representative: S1 the worst scenario did happen in the past; S2 is the worst scenario could occur at present; S3 and S4 are two typical future scenarios based on S2 considering TCI and SLR. Those justifications are added in the revised manuscript.

––––––––––––––––––––––––––––––––––

**Comment:** *[Conclusions] This section needs to be modified after implementing previous recommended changes.*

**Reply:** Yes, the section of conclusions with modifications considering the previous changes, especially those correspond to the methodology and whether the scenarios

are representative included is presented in the revised manuscript.
* * *
**Comment:** *Figure 10 is not needed.*

**Reply:** Figure 10 is the remotely sensed maps of the hardest three hit regions of storm tide inundation, which shows that the downtown of Cangnan city and Aojiang city, Longwan and Dongtou districts of Wenzhou city are of high risks of storm tide inundation considering TCI and SLR. It is believed that those results may knoll alarm clock for the local risk management of future potential storm tide inundation.
* * *